# Forgetting, Ignorance or Myopia: Revisiting Key Challenges in Online Continual Learning

**Xinrui Wang**[1,2]    **Chuanxing Geng**[1,2]    **Wenhai Wan**[3]    **Shao-yuan Li**[1,2]    **Songcan Chen**[1,2†]

[1]College of Computer Science and Technology, Nanjing University of Aeronautics and Astronautics
[2]MIIT Key Laboratory of Pattern Analysis and Machine Intelligence
[3] School of Computer Science and Technology, Huazhong University of Science and Technology

## Abstract

Online continual learning (OCL) requires the models to learn from constant, endless streams of data. While significant efforts have been made in this field, most were focused on mitigating the *catastrophic forgetting* issue to achieve better classification ability, at the cost of a much heavier training workload. They overlooked that in real-world scenarios, e.g., in high-speed data stream environments, data do not pause to accommodate slow models. In this paper, we emphasize that *model throughput*– defined as the maximum number of training samples that a model can process within a unit of time – is equally important. It directly limits how much data a model can utilize and presents a challenging dilemma for current methods. With this understanding, we revisit key challenges in OCL from both empirical and theoretical perspectives, highlighting two critical issues beyond the well-documented catastrophic forgetting: (i) Model's ignorance: the single-pass nature of OCL challenges models to learn effective features within constrained training time and storage capacity, leading to a trade-off between effective learning and model throughput; (ii) Model's myopia: the local learning nature of OCL on the current task leads the model to adopt overly simplified, task-specific features and *excessively sparse classifier*, resulting in the gap between the optimal solution for the current task and the global objective. To tackle these issues, we propose the Non-sparse Classifier Evolution framework (NsCE) to facilitate effective global discriminative feature learning with minimal time cost. NsCE integrates non-sparse maximum separation regularization and targeted experience replay techniques with the help of pre-trained models, enabling rapid acquisition of new globally discriminative features. Extensive experiments demonstrate the substantial improvements of our framework in performance, throughput and real-world practicality.

## 1   Introduction

Online continual learning (OCL) is the learning paradigm that enables models to learn continuously from a dynamic data stream $\mathfrak{D} = \{\mathcal{D}_1, \mathcal{D}_2, \ldots, \mathcal{D}_t, \ldots\}$, where $\mathcal{D}_t = \{x_i, y_i\}_{i=1}^{N_t}$ is the dataset of task $t$ sampled from distribution $\mu_t$. Existing OCL methods are designed to promote effective learning by mitigating catastrophic forgetting and improving model plasticity through various techniques like gradient regularization[41], contrastive learning[31], experience replay[16, 10] and knowledge distillation[1]. However, these methods often fail to consider the assessment of model throughput, an essential metric especially crucial for managing data streams with varying arrival speeds.

According to [1, 81], a *model's throughput* is defined as the maximum number of training sample data that a model can process within a unit of time. When the training speed of the model is slower than the speed that data stream arrives, the model is forced to discard some training data which wastes data

---

[†]Corresponding author. Code link: https://github.com/wxr99/Forgetting-Ignorance-or-Myopia-Revisiting-Key-Challenges-in-Online-Continual-Learning

and harms model's performance. Furthermore, maintaining a real-time accessible memory buffer, as required by current OCL methods, also proves challenging in real-world applications[38, 42]. Under these practical concerns, a reexamination of the challenges in OCL from both empirical and theoretical perspectives is in urgent need. In this paper, we reveal that **model's ignorance** caused by single pass nature of data streams and **model's myopia** resulting from the continuous arrival of tasks may be more impactful than the well-documented catastrophic forgetting phenomenon.

**Model's ignorance.** Our first focus lies in whether models can acquire sufficient discriminative features within the limited time during the single-pass training. To independently study this challenge and avoid other issues like catastrophic forgetting and task interference or collision caused by the continuous arrival of tasks, we introduce a relaxed setting for streaming data called **single task setting**. The data stream is sampled from a unified task, allowing data from any category to appear at any moment with equal probability. Under this controlled setting, we have observed that, 1) models trained from scratch under-perform significantly compared to expectations. The single-pass nature of OCL inhibits the model's ability to fully harness the semantic information from the data stream, a phenomenon we term as **model's ignorance**; 2) It is also noticed that existing strategies like contrastive learning and knowledge distillation to mitigate this issue significantly increase the training time of the model, consequently reducing its throughput.

**Model's myopia.** Even if models can quickly achieve decent performance on individual tasks, performance degradation remains a persistent issue for existing OCL models. Previous studies often attribute this to the model forgetting previously learned information. But in this paper, we propose a different perspective. We observed that, in OCL training, the model initially acquires perfect classification accuracy for a specific class, e.g., "car". However, there arrives a critical moment when the model becomes completely confused, mistakenly identifying a "car" as a newly introduced class (e.g., "truck"). We believe this confusion cannot solely be attributed to the model forgetting previous learned knowledge, as *the forgetting process should be gradual rather than abrupt*. Besides, as training progresses, the parameters in the final layer of the model's classifier become increasingly sparse. The emergence of such an *excessively sparse classifier* causes the model to focus on few discriminative features specialized for the current task. When the model is exposed to only a limited range of categories, this narrow focus on the current task restricts its capability to acquire features with broader discriminative power. We term this limitation **model's myopia**.

In addition to empirical verification, we adopt the Pac-Bayes theorem to provide insights into the dilemma between effective learning and model throughput. The upper bound of expected risk summation can be segmented into three terms correlated to empirical risk, model throughput and task divergence respectively. Our theory places a particular emphasis on model throughput, which has been long overlooked in the context of OCL. To the best of our knowledge, this is the first attempt to provide theoretical insights into the relationship between model throughput and performance in this area. Given that model needs to adapt to varying data flow rates to ensure its performance, this factor also warrants recognition in theoretical discussions. Plus, interestingly, model's myopia and forgetting can be perceived as two complementary aspects of the proposed task divergence term.

Built on the above analysis, we propose a new OCL framework called Non-sparse Classifier Evolution (NsCE), which capitalizes on the benefits of pre-trained initialization. This framework introduces a non-sparse regularization term and employs a maximum separation criterion between classes, aimed at mitigating the issue of parameter sparsity while maintaining the model's ability in differentiating classes. As a regularization applied uniformly across tasks, it helps to minimize the differences in the distributions of model parameters between tasks. Furthermore, to enhance the model's throughput and diminish the reliance on a real-time memory buffer, we propose an efficient selective replay mechanism. By selectively replaying past experiences, we specifically target data from classes that the model frequently misidentifies and implement the targeted binary classification tasks on them during experience replay. This approach not only enhances model's throughput but also boosts its performance in handling high-speed data streams. Extensive experiments demonstrate that these techniques are crucial for deploying a OCL model where high performance, real-world practicality and computational efficiency are all paramount.

## 2 Ignorance: Trade-off between Effective Learning and Model Throughput

In OCL, the most extensively studied challenge is the issue of catastrophic forgetting when learning new tasks. However, the poor performance of existing OCL methods on some large datasets [68]

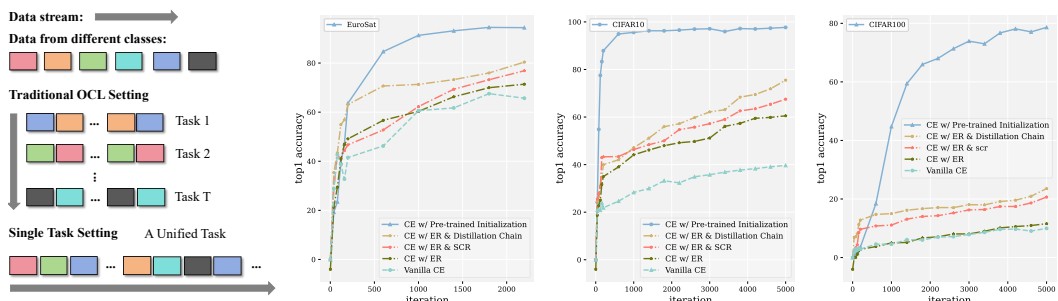

Figure 1: Real-time accuracy of OCL models trained under the standard cross entropy loss $L_{ce}$ both with and without pre-trained models (pre-trained on ImageNet) under our designed **single task setting** and the impact of some commomly used strategies[16, 49, 49]. Results on additional datasets, influence of different pre-trained models (pre-trained on different datasets, using different backbones and different pre-train tasks) and implementation details are provided in Appendix C.3.

inevitably leads us to question whether, before considering the issue of forgetting, the model can truly acquire sufficient discriminative features and subsequent classification capabilities. To isolate the challenge brought by the continual arrival of tasks and concentrate on model's behavior on the single pass data stream, we first construct a data stream setting called **single task setting**. As displayed in Figure1, we consolidate classes from multiple tasks into a single unified task, where samples from different classes are introduced at random timestamps with equal probability. It ensures a stable and balanced data stream, mitigating concerns about catastrophic forgetting or class imbalance.

**Unsatisfactory model performance.** Under this controlled setup, a simple supervised model is optimized using cross-entropy loss $\mathcal{L}_{ce} = -\sum_{i=1}^{N}\sum_{c=1}^{C} y_{i,c} \log(\phi(f_c(x_i)))$ where $\phi(\cdot)$ represents the classifier, $f(\cdot)$ denotes the feature extractor. We implemented common OCL strategies such as experience replay[16], contrastive learning[49, 64] and knowledge distillation[1] to make a comparison. As illustrated in Figure 1, models trained from scratch fail to reach satisfactory performance in single-pass training scenarios, unlike those benefiting from pre-trained initialization. On CIFAR100, model's average accuracy remains below 10% even without any inter-task interference. Single-pass nature of OCL prevents the model from fully leveraging the semantic information from the data stream, which we refer as **model's ignorance**. Meanwhile, as we integrate additional techniques like experience replay, contrastive learning, and distillation chains, the model's accuracy progressively improves, from 10% to 20%. Under such circumstances, leveraging additional prior knowledge seems to be the only viable solution. Empirical evidence also supports this viewpoint, as displayed by the performance when using pre-trained initialization in Figure1. While the selection of an appropriate pre-trained model is beyond the primary focus of this paper, we investigate the effects of various pre-training methods on different OCL downstream tasks in Appendix C.3.

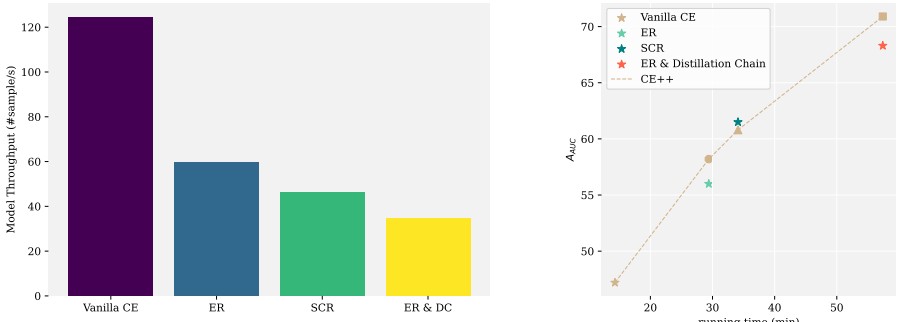

Figure 2: **Left:** Throughput of the model trained using vanilla cross-entropy, experience replay[16], supervised contrastive replay[49] and distillation chain[1]. **Right:** Performance($A_{AUC}$: Area Under the Curve of Accuracy) and running time of the above strategies on CIFAR10. "*CE++*" denotes that we compute and perform extra gradient descent per time step to match the delay of the compared-against strategies. All experiments are conducted under **single task setting**.

**Decreased model throughput.** Despite the partial effectiveness of commonly used strategies in mitigating **model's ignorance**, they inevitably extend the training time for the same amount of data and increase the demands on the memory buffer's real-time accessibility. As shown in Figure 2(**Left**), the integration of additional techniques consistently increases the training time. But in the context of OCL, this extended training duration leads to processing fewer data units per unit of time, resulting in lower *model throughput*. Given that the volume of training data is widely recognized as a crucial factor in determining model performance, this highlights a significant flaw in the current evaluation of OCL models. When the data stream's flow rate exceeds the training speed, model throughput and effective learning emerge as two interrelated factors subject to trade-offs.

More surprisingly, our findings suggest that these strategies are actually no more effective than simply training the model multiple times (*CE++*) on the same data to offset the delays caused by these methods. As illustrated in Figure 2, when handling data at the same flow rate, *CE++* can achieve comparable model performance through the application of experience replay, contrastive techniques and distillation chains. Furthermore, the challenges of maintaining a continuously accessible real-time memory buffer, caused by issues like network connectivity and privacy protection, are frequently overlooked, adding further obstacles to the effective implementation of these strategies.

## 3  Myopia: Key Factor for Performance Degradation

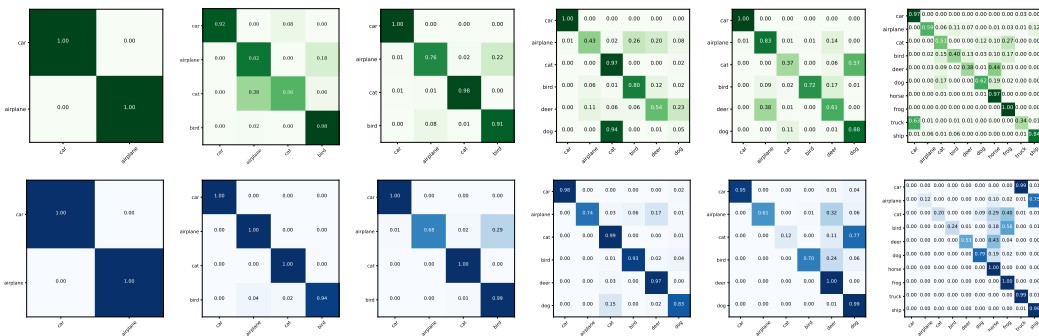

Figure 3: Normalized confusion matrix of NCM classifier (green) and softmax classifier (CIFAR10) (blue) with ImageNet supervised pre-trained initialization. Due to space limitations, we present a partial training process in the main text. Comprehensive training process is in AppendixE.

In addition to issues of **model's ignorance** and decreased model throughput, we recognize that while pre-trained initialization allows the model to quickly classify training data, relying solely on this initialization does not ensure satisfactory overall performance. Performance degradation continues to pose challenges in OCL [68]. Most previous studies attribute it to the phenomenon of catastrophic forgetting, which is caused by interference between current task and previously learned knowledge [66]. Some studies also suggest it is due to the model learning some trivial features[31, 68]. To clarify this issue, we conduct a comprehensive monitoring of the model's predictions by visualizing the predicting confusion matrix throughout the entire training process. Specifically, the model is trained by vanilla cross-entropy loss (Eq.1), without employing any other techniques.

$$\mathcal{L}_{ce} = -\sum_{t=1}^{T}\sum_{i=1}^{N_t}\sum_{c=1}^{C_t} y_i^{t,c}\log(\phi^c(f(x_i^t))), (x_i^t, y_i^t) \sim \mu_t. \tag{1}$$

We separately assess the discrimination ability of the classifier and feature extractor by evaluating the model's performance using a softmax classifier[16] and an online updating NCM (Nearest Class Mean) classifier[58] which are both very common approaches for prediction in OCL. For the NCM classifier, we compute a dynamic class mean prototype for each class using Equation 2 and apply a momentum update with the parameter $\lambda$. This update calculates the new class mean prototype $\mu_c^{new}$ based on the $n_c$ data points from class $c$ in the current data stream and the previous prototypes $\mu_c^{old}$. Then, class label of new data can be assigned to the most similar prototype.

$$\mu_c^{new} = (1-\lambda)\mu_c^{old} + \lambda\frac{1}{n_c}\sum_i f(x_i) \cdot \mathbb{I}\{y_i = c\}, y^* = \underset{c=1...C}{\arg\min} ||f(x) - \mu_c||. \tag{2}$$

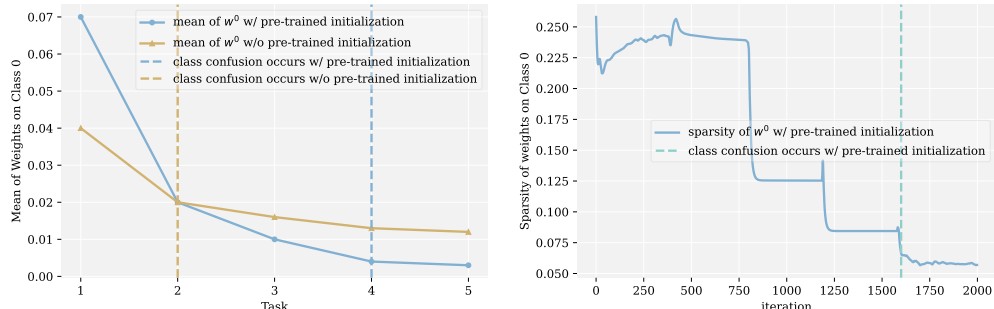

Figure 4: **Left:** averaged weights of the final FC layer for class 0 in CIFAR10. **Right:** $s(w)$ (lower $s(w)$ stands for increasing sparsity) of the final FC layer for $w^0$ corresponds to class 0 in CIFAR10. During the training of task 5, the class confusion occurs as Figure 3 where model classify "car" as "truck".

Although pre-trained initialization provides the model with a broader perspective and prior knowledge, performance degradation still occurs with the introduction of new tasks. As illustrated in Figure 3, the precision for the class 'car' dramatically drops from $0.95$ to nearly $0$ during the training of the fifth task. When closer examining the classes that the model confuses during training, such as 'car' and 'cat', we suppose that distinguishing between them becomes challenging when highly discriminative features from past tasks reappear in new classes. For example, 'car' and 'truck' share similar shapes and backgrounds, while 'cat' and 'dog' share similar textures, making differentiation difficult. Different from the well-documented catastrophic forgetting, we propose a different perspective that the confusion arises since the discriminative features or knowledge acquired from previous tasks may not be helpful in distinguishing these classes from some new classes in future tasks from the outset. In other words, the previously learned representations may not capture the essential characteristics necessary for effectively differentiating these classes with new classes and this is something perfectly normal even for us as humans. During the training process, the model naturally focuses on features and discriminant criteria that are more important for the current task. The independent arrival of tasks results in such **a myopic model**[70], which is the key factor in the decline of OCL performance.

Plus, we observed that softmax classifiers are more frequently susceptible to performance degradation compared to NCM classifiers. As depicted in Figure 3, during the fourth and fifth tasks, predictions made by the softmax classifier are more biased towards classes in the current task compared to those made by the NCM classifier. Meanwhile, although the features extracted by the model are initially separable, a biased classifier soon leads to significant confusion between classes, causing the features to lose discriminative power over time. Such phenomenons is better displayed by the complete visualization in Figure15 (AppednixE).

To better analyze the reasons behind model's performance degradation and verify our suppose on **model's myopia**, we evaluate and subsequently visualize the sparsity as $1/s(\cdot)$ and mean $m(\cdot)$ of the parameters in the final fully connected layer of the classifier for each task, as detailed in Eq.3. We represent this final fully connected layer by a matrix $W \in \mathcal{R}^{d \times C}$, where $d$ represents the feature dimension and $C$ denotes the number of classes. The variable $w^c$ refers to the $c$-th column vector extracted from $W$, and $w^c \in \mathcal{R}^d$.

$$m(w) = \sqrt{w_1^2 + w_2^2 + \cdots + w_d^2}; s(w) = \frac{(|w_1| + |w_2| + \cdots + |w_d|)/d}{\max(|w_1|, |w_2|, \cdots, |w_d|)} \qquad (3)$$

As depicted in Figure 4, the mean and sparsity of parameters in the classifier continuously decrease when each new task is introduced. Prior studies in continual learning [9, 37, 69] have linked this pronounced prediction bias towards recent tasks to the decreasing mean weights for old classes. Interestingly, unlike scenarios involving training from scratch, using a pre-trained initialization prevents the model from arbitrarily classifying class $0$ as belonging to the current task. Moreover, although the mean weights for the old classes consistently decline, class confusion only manifests with the arrival of task $4$. These observations all suggest that the reduction in weights is not solely responsible for the observed bias in the classifier. This leads us to question whether, during training

process, the model's criteria for categorizing become simpler. In other words, the model increasingly focuses solely on a limited set of discriminative features that it deems beneficial for the current task, and this focus is precisely the cause of **model's myopia**. This trend towards simplification is illustrated in Figure 4(**Right**), where there is a noticeable increase in the sparsity of parameters associated with older tasks as new ones are introduced. While relying on few discriminative features is beneficial in traditional supervised learning settings, in OCL, the emergence of such an *excessively sparse classifier* causes inevitable class confusions across tasks.

## 4 Theoretical Analysis

In addition to the empirical analysis, we try to provide theoretical insights into the OCL problem and illustrate the aforementioned trade-off between adequate feature learning and model throughput. Specifically, we approach the OCL problem from a Pac-Bayes perspective, as outlined in Theorem 4.1. Following the common notations used in Pac-Bayes literature [8], we define a model space $\mathcal{H}$ and a loss function $\ell : \mathcal{H} \times \mathcal{Z} \to \mathbb{R}^+$, which is bounded by a constant $K > 0$. Here, $\mathcal{Z}$ represents the whole training set and $\mu_t$ stands for data distribution of task $t$. In line with the approach described in [34], we denote a sequence of distributions $(Q_i)_{i=0..T}$ on $\mathcal{H}$, representing the evolution of the model's learning process. Here, $Q_i$ is the distribution of the model parameters after training task $i$, with $Q_0$ representing the initial parameter distribution and $T$ indicates the total number of tasks. Meanwhile, we denote the flow rate of the data stream (#samples coming from data stream in the unit time) as $v_s$, the model throughput (#samples model can train in the unit time) as $v_m$ and $\Delta_t$ as the duration time of task $t$ in the data stream. The sum of expected risks $\mathcal{R}$ across different tasks can be written as $\sum_{t=1}^T \mathbb{E}_{h_t \sim Q_t} \left[ \mathbb{E}_{z_t \sim \mu_t} [\ell(h_t, z_t)] \right]$ and the empirical risk $\hat{\mathcal{R}}$ is $\sum_{t=1}^T \sum_{j=1}^{m_t} \frac{\mathbb{E}_{h_t \sim Q_t} \left[ \ell(h_t, z_j^t) \right]}{m_t}$.

**Theorem 4.1.** *For any distributions $\mu_1, ..., \mu_T$ over $\mathcal{Z}$, let $\mathcal{D}_t$ be an iid set with $m_t = \min(v_s, v_m)\Delta_t$ samples sampled from $\mu_t$ as the dataset of task $t$, for any $\lambda > 0$ and any online predictive sequence $(Q_0, Q_1, ..., Q_T)$, the following inequality holds with probability $1 - \delta$:*

$$\mathcal{R} \leq \hat{\mathcal{R}} + \underbrace{\sum_{t=1}^T \frac{\lambda K^2}{\min(v_s, v_m)\Delta_t}}_{\mathcal{M}} + \underbrace{\sum_{t=1}^T \frac{\mathrm{KL}(Q_t \| Q_{t-1})}{\lambda}}_{\mathcal{D}} + \underbrace{\frac{T \log(T/\delta)}{\lambda}}_{constant}. \tag{4}$$

It is clear that the upper bound of $\mathcal{R}$ can be segmented into three terms, along with a constant related to the task number $T$. They are identified as empirical risk $\hat{\mathcal{R}}$, model throughput term $\mathcal{M}$, and task divergence term $\mathcal{D}$. Among them, the model throughput term $\mathcal{M}$ is determined by the amount of data accessible to the model and this is directly influenced by the model's throughput $v_m$ when data stream's flow rate $v_s$ exceeds $v_m$. However, achieving a lower empirical risk $\hat{\mathcal{R}}$ by data augmentations, knowledge distillation or training the model for multiple times typically requires more training time and consequently reducing model throughput $v_m$, which theoretically explains the trade-off we mentioned in Section 2 between $\hat{\mathcal{R}}$ and $\mathcal{M}$.

When examining the task divergence term $\mathcal{D}$ more closely, we see that nearly all OCL methods aiming at addressing forgetting attempt to minimize the deviation from the parameter distribution of previous tasks by adjusting the current model parameter distribution. Our proposed concept of **model's myopia** offers a fresh perspective. By aligning the distribution of current model with future ones, the divergence term $\mathcal{D}$ can be also reduced. It leads us to considering the use of structural constraints (e.g. non-sparse regularization) or pre-trained initialization as promising strategies to enhance model performance. Meanwhile, it's important to acknowledge that our analysis has limitations; the bound discussed is intended to illustrate the sum of generalization risk for each individual task and can not represent a global expectation. We provide detailed discussions and proof in Appendix B.

## 5 Method

After conducting a series of analysis on the key challenges in OCL, in this section, we provide the NsCE framework to tackle these issues based on the utilization of pre-trained initialization. Our goal is not only to reduce the risks of model ignorance and myopia but also to enhance the throughput of OCL models. We aim to accelerate training speeds to keep pace with data stream progress and reduce the dependence of existing methods on real-time accessible memory buffers.

**Non-sparse regularization.** Unlike some previous methods that aim to acquire task-specific features capable of generalization, in this study, we acknowledge the unrealistic expectation of obtaining a model with absolute discriminative ability within a limited scope of classes, even with the rich prior knowledge provided by a pre-trained model. Instead, our focus lies in ensuring the diversity of discriminant features during training and enabling the model to swiftly develop the ability to differentiate between categories from different tasks. As posited in the previous section, in the context of OCL, contrary to traditional settings where a sparse classifier is often considered desirable for achieving high classification performance[22, 46], the overly sparse parameters can cause the model to focus solely on a limited set of highly discriminative features, increasing the risk of **model's myopia**. To mitigate this issue, we propose a straightforward idea of constraining the sparsity of the final fully connected layer of (softmax classifier) of the model. Our goal is to ensure that the model maintains a diverse set of discriminative features during training, allowing it to effectively handle different tasks without being overly biased towards specific features. Meanwhile, it helps reduce the divergence of model distribution across tasks. In specific implementation, considering that the $\max(\cdot)$ function is easily affected by a small number of outliers in the parameters, we opt to replace it with the $l_2$ norm as a more robust alternative in our sparsity regularization:

$$\mathcal{L}_s = -\sum_{c=1}^{C} \frac{(|w_1^c| + |w_2^c| + \cdots + |w_d^c|)/d}{\sqrt{w_1^{c2} + w_2^{c2} + \cdots + w_d^{c2}}}. \tag{5}$$

**Maximum separation.** While a smooth classifier can help to mitigate the model's myopia, it hinders the model's ability to rapidly perform classification in the current task. Moreover, in OCL, it is also hard to have simultaneous access to data from all categories, especially when there are restrictions on the use of memory buffers. This leads to severe class imbalance during the learning process, which is a well-recognized challenge in the context of continual learning[72, 76, 39]. Thus, we draw inspiration from the famous Neural Collapse [55] and Maximum Class Separation criterion[39]. It also serves as a structure constraints on model parameters to minimize the model distribution divergence across the tasks. For learned representations from different categories $\{f(x_1), f(x_2), \cdots f(x_{C_t})\}$, their cosine similarity should satisfy a maximum separation criterion and converge to an ideal simplex equiangular tight frame (ETF), $\forall_{i,j,i \neq j} \langle f(x_i), f(x_j) \rangle = -\frac{1}{C_t - 1}$.

$$\mathcal{L}_p = \frac{1}{C_t^2} \sum_{i,j=1}^{C} (\langle f(x_i), f(x_j) \rangle - p_{ij})^2, \quad p_{ij} = \frac{C_t}{C_t - 1} \delta_{i,j} - \frac{1}{C_t - 1} \tag{6}$$

where $\delta_{i,j}$ is Kronecker delta symbol that designates the number 1 if $i = j$ and 0 if $i \neq j$. To address categories not present in the current task, we use the class mean in Eq.2 to replace the representation of the corresponding category. Thus, we can denote our total loss function as:

$$\mathcal{L} = \mathcal{L}_{ce} + \gamma(\mathcal{L}_p + \mathcal{L}_s) \tag{7}$$

**Targeted experience replay.** To enable the model to learn globally discriminative features and correct existing class confusion, we prioritize the categories that the model has previously struggled to distinguish when accessing the memory buffer. During experience replay, we compute a confusion matrix to identify frequently confused categories. To address each group of confused categories, we devise a separate binary classification loss specifically designed to expedite the acquisition of discriminative abilities between these confused classes, as shown in Eq.8.

$$\mathcal{L}_b = -\sum_{i=1}^{|\mathcal{B}|} \sum_{m,n=1}^{C} \mathbb{I}\{\mathcal{C}_{m,n}^b > \tau\} \cdot [y_i^m \log(\phi^m(f(x_i))) + y_i^n \log(\phi^n(f(x_i)))], m \neq n. \tag{8}$$

$\mathcal{C}^b$ represents a normalized confusion matrix and $\mathcal{C}_{m,n}^b > \tau$ indicates that the proportion of data belonging to class $m$ being classified as class $n$ exceeds the threshold $\tau$. To further enhance model's throughput and diminish the reliance on a real-time memory buffer, we impose limitations on the number of requests allowed to retrieve data from the memory buffer. Compared to traditional experience replay, our way of replay achieves higher model throughput and is more specifically targeted at addressing existing class confusion. For a complete description of the algorithm process, please refer to Algorithm1. Moreover, our findings indicate that when selecting an appropriate pre-trained model, halting gradient back-propagation in the feature extractor often enhances throughput without compromising performance. More discussions are provided in AppendixC.2.2.

Table 1: Best $A_{AUC}$ is highlighted in **bold**, second best is shown underlined.

| Method | CIFAR-10 | | | CIFAR-100 | | | EuroSat | | |
|---|---|---|---|---|---|---|---|---|---|
| | $M=0.1k$ $Freq=1/100$ | $M=0.2k$ $Freq=1/50$ | $M=0.5k$ $Freq=1/10$ | $M=0.5k$ $Freq=1/100$ | $M=1k$ $Freq=1/50$ | $M=2k$ $Freq=1/10$ | $M=0.1k$ $Freq=1/100$ | $M=0.2k$ $Freq=1/50$ | $M=0.5k$ $Freq=1/10$ |
| iCaRL[58] | 80.6±0.5 | 83.9±0.4 | 88.2±0.4 | 55.1±0.2 | 57.9±0.4 | 67.7±0.2 | 58.7±0.4 | 75.2±1.1 | 80.4±0.7 |
| EWC[41] | 81.7±0.7 | 85.5±1.2 | 91.2±0.7 | 60.7±0.8 | 62.9±3.2 | 67.2±1.1 | 61.0±0.8 | 72.6±0.5 | 83.9±1.1 |
| DER++[10] | 81.5±1.2 | 86.7±0.8 | 89.9±1.0 | 59.2±0.9 | 61.1±0.8 | 69.2±1.4 | 45.0±6.0 | 78.2±2.4 | 81.9±2.1 |
| PASS[82] | 82.0±0.8 | 85.2±0.6 | 90.3±1.2 | 61.2±1.3 | 62.9±0.9 | 67.0±1.5 | 50.1±3.1 | 78.1±1.2 | 83.5±0.8 |
| MC-SGD w/ SAM[51] | 81.8±0.5 | 83.9±1.2 | 90.5±0.4 | 60.2±0.8 | 63.1±0.8 | 71.8±0.8 | 61.3±0.9 | 78.9±0.5 | 84.6±1.0 |
| AGEM[15] | 78.6±0.7 | 81.2±1.1 | 85.7±0.9 | 50.2±0.7 | 58.9±1.1 | 67.4±0.8 | 56.4±0.7 | 67.7±0.9 | 81.9±1.0 |
| ER[16] | 82.6±0.5 | 85.4±0.4 | **91.2±0.2** | 61.3±0.3 | 64.6±0.4 | 71.2±1.2 | 58.6±0.8 | 70.5±0.6 | 84.0±0.8 |
| MIR[4] | 82.4±0.4 | 85.7±0.7 | 89.9±1.0 | 62.9±0.6 | 63.3±0.7 | 71.2±1.4 | 59.0±1.0 | 71.1±0.9 | 84.2±1.1 |
| ASER[60] | 80.7±1.2 | 84.4±0.6 | 87.2±1.0 | 60.1±0.8 | 62.3±0.9 | 70.9±1.4 | 59.4±1.0 | 72.0±0.8 | 83.7±0.4 |
| SCR[49] | 83.8±0.2 | 85.9±1.4 | 90.5±0.9 | 61.5±0.4 | 62.7±0.2 | 71.2±0.4 | 52.1±0.8 | 75.9±0.7 | 84.8±0.6 |
| DVC w/o Aug[29] | 80.5±0.2 | 85.9±0.3 | 89.2±0.7 | 57.9±0.6 | 58.9±0.6 | 67.4±0.8 | 52.0±0.9 | 69.1±0.8 | 82.7±1.1 |
| DVC[29] | 81.1±0.2 | 85.8±0.4 | 90.3±0.5 | 61.6±0.8 | 62.9±1.0 | 70.7±0.9 | 53.6±0.7 | 72.7±1.1 | 85.3±1.0 |
| OCM w/o Aug[31] | 79.1±1.5 | 83.3±1.4 | 90.1±2.0 | 60.9±0.8 | 58.4±1.6 | 69.5±0.4 | 46.7±1.2 | 78.6±1.0 | 83.1±0.4 |
| OCM w/ Aug[31] | 82.1±2.9 | 85.2±2.1 | 90.2±2.7 | 61.3±1.5 | 60.3±1.1 | 70.2±0.7 | 51.9±1.4 | 76.8±1.2 | 84.0±0.9 |
| OnPro w/ Aug[68] | 81.1±0.6 | 86.1±0.7 | 90.1±0.8 | 62.9±0.7 | 63.7±0.8 | 70.5±0.9 | 52.8±6.8 | 75.2±1.0 | 83.7±0.9 |
| NsCE | **89.9±0.4** | **90.4±0.4** | 90.7±1.0 | **74.1±0.7** | **75.5±0.8** | **79.7±0.9** | **75.7±0.4** | **83.4±0.7** | **86.3±0.4** |

# 6 Experiments

Before delving into the specifics of experimental results, we highlight a significant distinction in the utilization of memory buffers between our study and other works. In this work, we impose limitations on the number of requests allowed to retrieve data from the memory buffer. We evaluate our models on six image datasets, incorporating realistic task overlaps to mimic practical scenarios[43]. Training is harmonized using ViT architectures and the AdamW optimizer, with consistent training batch size and initialization (MAE pre-training for ImageNet, supervised pre-training for others). We compare our NsCE method 8 replay based methods and 5 regularization based ones. To more effectively assess model performance over time, we employ $A_{AUC}$ as the primary metrics. Specifically, we evaluate the model's accuracy on all previously encountered tasks at intervals of every 100 training iterations and use these data points to calculate $A_{AUC}$. Due to limited space, we leave detailed descriptions of the implementation, evaluation metrics, datasets and comparison methods in AppendixC.1.

Table 2: $A_{AUC}$ on on large-scale real-world online domain-incremental data stream. We discard OCM on ImageNet due to its significantly higher runtime and computational memory costs.

| Method | CLEAR-10 | | CLEAR-100 | | ImageNet |
|---|---|---|---|---|---|
| | $M=0.1k$ $Freq=1/100$ | $M=0.2k$ $Freq=1/50$ | $M=1k$ $Freq=1/100$ | $M=2k$ $Freq=1/50$ | $M=10k$ $Freq=1/500$ |
| ER | 87.3±1.0 | 87.9±0.5 | 80.1±0.6 | 82.0±0.8 | 55.6±0.4 |
| DER++ | 87.4±0.5 | 88.1±0.9 | 78.5±1.1 | 80.4±0.6 | 46.5±0.4 |
| EWC | 88.0±0.4 | 89.0±0.2 | 81.1±0.3 | 81.2±0.5 | 50.9±1.0 |
| iCarl | 86.2±0.8 | 87.1±1.1 | 77.1±0.8 | 80.4±0.9 | 57.1±1.8 |
| SCR | 84.1±0.7 | 85.9±0.6 | 67.2±0.7 | 79.7±0.4 | 52.5±2.4 |
| OCM | 88.1±0.5 | 89.2±0.6 | 80.0±0.9 | 82.5±1.0 | ××××× |
| DVC | 86.4±0.3 | 87.0±0.7 | 79.4±0.2 | 80.2±0.7 | 53.1±0.6 |
| OnPro | 86.9±1.4 | 88.1±0.9 | 80.4±0.3 | 82.3±0.2 | 52.2±0.4 |
| NSCE | **89.2±0.7** | **91.4±0.8** | **84.3±0.4** | **85.7±0.3** | **61.6±0.7** |

**Main Results and Analysis.** We conduct a comprehensive evaluation of our method by comparing its performance with several existing state-of-the-art OCL methods as well as various continual learning variants. Table1 displays the $A_{AUC}$ (area under the accuracy curve) for three synthetic benchmark datasets, showcasing the impact of different memory buffer sizes and replay frequencies. This evaluation metric offers a more comprehensive assessment compared to the commonly used average accuracy[43]. The results demonstrate that our proposed method, NsCE, consistently outperforms other approaches. Notably, the performance improvement achieved by NsCE is particularly significant when the memory buffer size is relatively small and the number of memory buffer access times is very limited. This finding highlights that the proposed framework helps prevent the model from excessively focusing on the current task which is crucial when memory capacity or access frequency are constrained. Plus, we also evaluate NsCE on real-world domain incremental datasets large-scale image classification dataset. It allows us to assess the performance and generalizability of our approach in real-world scenarios, where the challenges and characteristics may differ. Table2 demonstrates that NsCE can also enhance the performance in real-world domain incremental settings and complex data streams. More experimental results including model's last time accuracy, sensitivity analysis and evaluation on model throughput are provided in C.2. Moreover, we visualize the predictions of our proposed NsCE in Figure5. From the visualization, it is evident that our model quickly learns the current task while avoiding confusion between past categories and categories in the current task as much as possible, effectively alleviating **model's ignorance and myopia**.

**Ablation Studies.** To investigate the specific effects of different proposed components, we conduct a series of ablation studies. From Table3, we can draw several observations: (1) Each component we

Table 3: Ablation study of the proposed NsCE framework.

| Method | CIFAR10 $M = 0.1k$ $Freq = 1/100$ | CIFAR100 $M = 0.5k$ $Freq = 1/100$ | EuroSat $M = 0.1k$ $Freq = 1/100$ | CLEAR100 $M = 1k$ $Freq = 1/100$ | ImageNet $M = 10k$ $Freq = 1/500$ |
|---|---|---|---|---|---|
| vanilla $\mathcal{L}_{ce}$ w/ ER | 82.6±0.5 | 61.3±0.3 | 58.6±0.8 | 80.1±0.6 | 55.6±0.4 |
| vanilla $\mathcal{L}_{ce}$ w/ ER & $\mathcal{L}_s$ | 84.5±1.3 | 64.5±0.7 | 62.0±0.4 | 77.6±0.8 | 56.2±0.7 |
| vanilla $\mathcal{L}_{ce}$ w/ ER & $\mathcal{L}_s$ & $\mathcal{L}_p$ | 86.2±0.8 | 66.1±0.4 | 66.9±0.7 | 81.3±1.1 | 59.4±1.1 |
| vanilla $\mathcal{L}_{ce}$ w/ targeted ER | 87.2±0.9 | 71.9±0.9 | 72.4±0.4 | 83.7±0.6 | 58.2±1.3 |
| NsCE | **89.9±0.4** | **74.1±0.7** | **75.7±0.4** | **84.3±0.4** | **61.6±0.7** |

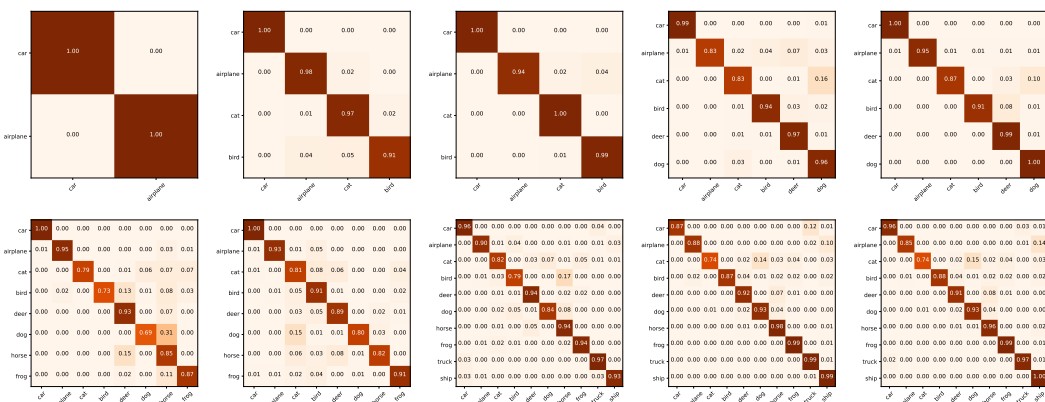

Figure 5: The detailed normalized confusion matrix (CIFAR10) evolution of our proposed NsCE framework (memory buffer size is 100 and replay frequency is 1/100).

propose provides performance improvements, among which **targeted ER** has the most obvious effect. (2) Constraints on classifier sparsity, as defined by $\mathcal{L}_s$, prove to be more effective in class incremental scenarios where model's myopia tends to be more pronounced. (3) The maximum separation term $\mathcal{L}_p$ achieves consistent performance improvements across datasets.

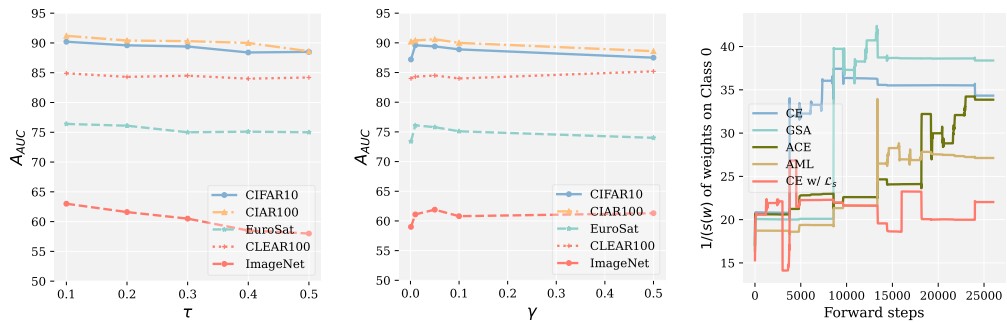

Figure 6: **Left:** Sensitivity analysis on $\tau$ and $\gamma$. **Right:** Sparsity ($1/s(w)$) of the classifier under different algorithms.

**Sensitivity analysis.** We analyze the impact of the threshold $\tau$ in targeted experience replay and the coefficient on the non-sparse maximum separate regularization. As depicted in Figure 6, we observe that as the threshold $\tau$ increases, our proposed NsCE has a relatively lower area under the accuracy curve ($A_{AUC}$). This trade-off between performance and efficiency is expected, as higher values of $\tau$ lead to fewer samples being replayed, resulting in improved model throughput but potentially compromising performance. Furthermore, our approach demonstrates robust outcomes when the coefficient $\gamma$ is not too small, and it basically achieves the best performance when $\gamma = 0.01$.

**Classifier sparsity.** We are also very interested in how sparsity would be affected by the proposed NsCE and methods focusing on re-arranged last layer weight updates. After implementing ER-ACE and ER-AML[2, 12], we found that the phenomenon of parameters rapidly becoming sparse is indeed somewhat mitigated, though not as significantly as with our proposed regularization term $\mathcal{L}_s$, as illustrated in Figure 6. While incorporating ACE or AML can also boost performance for baselines

like ER and SCR. We believe that when ACE and AML nudge the learned representations to be more robust to new future classes, they indirectly decrease the sparsity of the model parameters. For GSA[32], the sparsity is not affected. But we are not entirely sure whether this part is perfectly embedded or if further tuning would help, as the authors only provide hyperparameters for CIFAR-100. For SS-IL, we did not find its implementation, so it may be left for future works.

## 7 Conclusion

In this study, we conduct a thorough reevaluation of the major challenges in current OCL methods. We delve into the underlying causes of these challenges and the limitation of existing methods. Our analysis highlights two critical limiting factors: **model's ignorance and myopia**, which can have a more significant impact than the widely recognized issue of catastrophic forgetting. Furthermore, we introduce the NsCE framework, which incorporates non-sparse maximum separation regularization and targeted experience replay techniques with a focus on balancing performance, throughput and practicality. Our work aims to provide a fresh perspective and inspire the OCL field to prioritize both model's performance and efficiency in more real-world scenarios.

## 8 Acknowledgments and Disclosure of Funding

This work was supported by the National Science and Technology Major Project 2022ZD0114801, Natural Science Foundation of China (NSFC) (Grant No.62376126), the National Key R&D Program of China (2022ZD0114801), National Natural Science Foundation of China (61906089), Natural Science Foundation of China (NSFC) (Grant No.62106102), Natural Science Foundation of Jiangsu Province (BK20210292), Graduate Research and Practical Innovation Program at Nanjing University of Aeronautics and Astronautics (xcxjh20221601).

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

# A  Related Works

**Continual Learning.** Continual learning is a research field dedicated to learning continuously while mitigating the forgetting of previously acquired knowledge[54, 52, 27, 52, 27, 63]. Most continual learning approaches employ three types of techniques. Regularization-based approaches introduce regularization terms or constraints to the learning process to preserve previously learned knowledge[41, 75, 3, 25]. Memory-based approaches utilize external memory buffers or replay mechanisms to store and replay past data, allowing the model to retain access to previous experiences[33, 82, 15, 16, 60, 65]. Architecture-based approaches involve modifying the model architecture to facilitate continual learning[61, 73, 50, 40]. Additionally, reducing storage overhead and minimizing dependence on hardware devices are issues of concern in the research community[67, 77].

**Online Continual Learning.** Online continual learning (OCL) serves as a more realistic extension of continual learning. Unlike traditional batch learning, where the entire dataset for each task is available upfront, OCL operates in scenarios where data distributions dynamically change over time. In OCL, similar to memory-based approaches in CL, most methods leverage a real-time accessible memory buffer and employ various experience replay methods to mitigate the issue of forgetting[15, 16, 4, 6, 60, 5, 19, 14, 58, 21, 62, 67, 11]. Besides, other OCL methods aim to improve the learning of better features and classifiers in a single-pass training manner[58]. Techniques like contrastive learning[49, 13], mutual information maximization[29, 31], and prototype learning[82, 68] have been employed to enhance the discriminative abilities of the model and improve its performance. Moreover, there are other works that focus a more proper evaluation of existing algorithms[43, 28]. Compared with methods that aim for better performance, we focus on rethinking key challenges in OCL and then design a framework under more realistic throughput and storage constraints. Despite that online learning has been extensively studied by theoretical community, research on generalization bounds tailored for online continual learning remains scarce and our bound serves a simple attempt to bridge the performance and model throughput.

**Online Continual Learning with Pre-trained Models.** The utilization of pre-trained models has become a common approach in various machine learning tasks, including transfer learning[74, 17], natural language processing[24] and class-incremental learning[51, 78]. While the effectiveness of pre-trained models has been well-established for these applications, only a few works[45] have explored their impact on OCL. These studies reveal underperforming algorithms can become very competitive when considering when using pre-trained models [35, 18, 57, 30].

# B  Detailed Proof and Limitations

We first reintroduce the classical PAC-Bayes adapted from [7, 8] as the Lemma.

**Lemma B.1** (Adapted from [8], Thm 4.1). *Let $\mathcal{D} = (z_1, ..., z_m)$ be an iid set sampled from the law $\mu$. For any data-free prior $P$, for any loss function $\ell$ bounded by $K$, any $\lambda > 0, \delta \in [0, 1]$, one has with probability $1 - \delta$ for any posterior $Q \in \mathcal{M}_1(\mathcal{H})$:*

$$\mathbb{E}_{h \sim Q} \mathbb{E}_{z \sim \mu}[\ell(h, z)] \leq \frac{1}{m} \sum_{i=1}^{m} \mathbb{E}_{h \sim Q}[\ell(h, z_i)] + \frac{\mathrm{KL}(Q\|P) + \log(1/\delta)}{\lambda} + \frac{\lambda K^2}{2m},$$

*where $\mathcal{M}_1(\mathcal{H})$ denotes the set of all probability distributions on $\mathcal{H}$.*

*Remark* B.2. Theorem B.1 is a special case of the original theorem from [8] as we take the case of a bounded loss which implies the subgaussianity of the random variables $\ell(., z_i)$ and then allows us to recover the factor $\frac{\lambda K^2}{m}$.

Following [59], we introduce the notion of *stochastic kernel* which formalise properly data-dependent measures within the PAC-Bayes framework. First, for a fixed predictor space $\mathcal{H}$, we set $\Sigma_{\mathcal{H}}$ to be the considered $\sigma$-algebra on $\mathcal{H}$. We denote $\mathcal{M}_1(\mathcal{H})$ as the set of all probability distributions on $\mathcal{H}$.

**Definition B.3** (Stochastic kernels[59]). A *stochastic kernel* from $\mathcal{D} = \mathcal{Z}^m$ to $\mathcal{H}$ is defined as a mapping $Q : \mathcal{Z}^m \times \Sigma_{\mathcal{H}} \to [0; 1]$ where

- For any $B \in \Sigma_{\mathcal{H}}$, the function $\mathcal{D} = (z_1, ..., z_m) \mapsto Q(\mathcal{D}, B)$ is measurable,
- For any $\mathcal{D} \in \mathcal{Z}^m$, the function $B \mapsto Q(\mathcal{D}, B)$ is a probability measure over $\mathcal{H}$.

We denote by $\text{Stoch}(\mathcal{D}, \mathcal{H})$ the set of all stochastic kernels from $\mathcal{S}$ to $\mathcal{H}$ and for a fixed $S$, we set $Q_{\mathcal{D}} := Q(\mathcal{D}, .)$ the data-dependent prior associated to the sample $S$ through $Q$.

Following Theorem B.3, we provide a formal definition of the **online predictive sequence** as [34]:

**Definition B.4.** A sequence of stochastic kernels $(P_i)_{i=1..m}$ is denoted as an ***online predictive sequence*** if (i) for all $i \geq 1, S \in \mathcal{Z}^m, P_i(\mathcal{D}, .)$ is $\mathcal{F}_{i-1}$ measurable and (ii) for all $i \geq 2, P_i(\mathcal{D}, .) \gg P_{i-1}(\mathcal{D}, .)$.

For $P, Q \in \mathcal{M}_1(\mathcal{H})$, the notation $P \ll Q$ indicates that $Q$ is absolutely continuous wrt $P$ (i.e. $Q(A) = 0$ if $P(A) = 0$ for measurable $A \subset \mathcal{H}$). Before giving a detailed proof, let us first reclaim our theorem.

**Theorem B.5.** *For any distributions* $\mu_1, ..., \mu_T$ *over* $\mathcal{Z}$, $\mathcal{D}_t = (z_1^t, ..., z_{m_t}^t)$ *an iid set with* $m_t = \min(v_s, v_m)\Delta_t$ *samples sampled from* $\mu_t$ *as the dataset of task* $t$ *and* $v_m$ *is the model throughput,* $v_s$ *is the flow rate of the stream and* $\Delta_t$ *is the the time of the data stream for task* $t$, *for any* $\lambda > 0$ *and any online predictive sequence (used as both priors and posteriors)* $(Q_0, Q_1, ..., Q_T)$, *the following inequality holds with probability* $1 - \delta$:

$$\sum_{t=1}^{T} \mathbb{E}_{h_t \sim Q_t}\left[\mathbb{E}_{z_t \sim \mu_t}[\ell(h_t, z_t)]\right] \leq \sum_{t=1}^{T}\sum_{j=1}^{m_t} \frac{\mathbb{E}_{h_t \sim Q_t}\left[\ell(h_t, z_j^t)\right]}{m_t} +$$

$$\sum_{t=1}^{T} \frac{\text{KL}(Q_t \| Q_{t-1})}{\lambda} + \sum_{t=1}^{T} \frac{\lambda K^2}{2m_t} + \frac{T \log(T/\delta)}{\lambda}.$$

*Proof.* For each task $t$ in OCL, we can consider $Q_{t-1}$ as a prior since it doesn't depend on the dataset $\mathcal{D}_t$. By applying Theorem B.1, we can have that let $\mathcal{D}_t = (z_1, ..., z_{m_t})$ be an iid set sampled from the law $\mu_t$. For data-free prior $Q_{t-1}$, for any loss function $\ell$ bounded by $K$, any $\lambda > 0, \tilde{\delta} \in [0, 1]$, one has with probability $1 - \tilde{\delta}$ for a data-dependent posterior $Q_t \in \mathcal{M}_1(\mathcal{H})$:

$$\mathbb{E}_{h_t \sim Q_t}\left[\mathbb{E}_{z_t \sim \mu_t}[\ell(h_t, z_t)]\right] \leq \sum_{j=1}^{m_t} \frac{\mathbb{E}_{h_t \sim Q_t}\left[\ell(h_t, z_j^t)\right]}{m_t} + \frac{\text{KL}(Q_t \| Q_{t-1})}{\lambda} + \frac{\lambda K^2}{2m_t} + \frac{\log(\tilde{\delta})}{\lambda}.$$

We then make $\tilde{\delta} = \delta/T$ and take an union bound on all tasks to ensure with probability $1 - \delta$ for any $t \in 1, 2, ...T$:

$$\mathbb{E}_{h_t \sim Q_t}\left[\mathbb{E}_{z_t \sim \mu_t}[\ell(h_t, z_t)]\right] \leq \sum_{j=1}^{m_t} \frac{\mathbb{E}_{h_t \sim Q_t}\left[\ell(h_t, z_j^t)\right]}{m_t} + \frac{\text{KL}(Q_t \| Q_{t-1})}{\lambda} + \frac{\lambda K^2}{2m_t} + \frac{\log(T/\delta)}{\lambda}.$$

Then, by taking a sum on all tasks, we can have the following result with probability $1 - \delta$:

$$\sum_{t=1}^{T} \mathbb{E}_{h_t \sim Q_t}\left[\mathbb{E}_{z_t \sim \mu_t}[\ell(h_t, z_t)]\right] \leq \sum_{t=1}^{T}\sum_{j=1}^{m_t} \frac{\mathbb{E}_{h_t \sim Q_t}\left[\ell(h_t, z_j^t)\right]}{m_t} + \sum_{t=1}^{T}\left[\frac{\text{KL}(Q_t \| Q_{t-1})}{\lambda} + \right.$$

$$\left. \frac{\lambda K^2}{2m_t} + \frac{\log(T/\delta)}{\lambda}\right]$$

$$= \underbrace{\sum_{t=1}^{T}\sum_{j=1}^{m_t} \frac{\mathbb{E}_{h_t \sim Q_t}\left[\ell(h_t, z_j^t)\right]}{m_t}}_{\hat{\mathcal{R}}} + \underbrace{\sum_{t=1}^{T} \frac{\lambda K^2}{2\min(v_s, v_m)\Delta_t}}_{\mathcal{M}}$$

$$+ \underbrace{\sum_{t=1}^{T} \frac{\text{KL}(Q_t \| Q_{t-1})}{\lambda}}_{\mathcal{D}} + \underbrace{\frac{T \log(T/\delta)}{\lambda}}_{const}.$$

$\square$

The flexibility of the classical PAC-Bayes bound allows the stochastic kernels $Q_t$ to be either data-dependent distributions or not, as stated in Theorem B.1. In the case of data-dependent distributions, the only available prior we can select in the predictive sequence $Q_t$ is the initial prior distribution $Q_0$. Meanwhile, it is possibly the largest term in the divergence term $D$. This emphasizes the significance of a good initialization and pre-trained model in achieving favorable results. Furthermore, by examining Lemma B.5, we can observe that the derived bound deteriorates as the number of tasks $T$ increases. This deterioration arises from the growing number of new tasks, which makes online continual learning more challenging. As the model needs to adapt and accommodate an expanding set of tasks, the learning process becomes increasingly complex and prone to performance degradation.

Furthermore, the third model throughput term in the bound emphasizes the significance of model throughput, as it directly impacts $v$. Many existing techniques in OCL, such as data augmentation, knowledge distillation and gradient constraints all increase the training time, consequently reducing the amount of data ($m_t$) that the model can process when the data stream's flow rate surpasses the $v$. A lower model throughput not only hampers the practicality of the OCL method but also restricts the model's generalization ability.

Compared to traditional generalization bounds that only consider the final output of an algorithm, the left side of Theorem B.5 evaluates the performance of the model at each time step. This distinction is crucial because in continual learning scenarios, the model's performance should be assessed and monitored throughout the learning process, rather than solely focusing on the final outcome. Indeed, considering the performance of the model at each time step can be seen as a compromise that aligns the generalization gap with a notion of regret. Compared with the regret bound provided in [34], the deteriorated convergence rate is mainly caused by the fact that at each time step we don't have an access to all the past data to predict the future as the projected Online Gradient Descent (OGD) algorithm. In summary, this is just a simple and natural extension of [8] in the context of OCL. However, we believe that Theorem B.5 already provides some theoretical guidance for the issues that OCL needs to address. In future work, we hope to obtain a more in-depth theoretical analysis specifically for this problem.

**Limitations.** First, it is important to clarify that the bound discussed is specifically used to illustrate the generalization risk associated with each individual task, rather than representing a global risk. This limitation means that our current theoretical analysis can not extend to data streams composed of disjoint tasks. Despite this constraint, we believe that it does not detract from the main findings presented in the core sections (**model's ignorance and myopia**) of our paper. Minimizing the distribution divergence across tasks remains one of the most fundamental concepts in OCL, despite our theoretical results can not fully reflect the benefits of mitigating model's forgetting or myopia. Part of the reason is that such a sum of risk is easier to account for the problem of the trade-off between effective learning and model throughput. On the other hand, it is actually extremely challenging to directly establish the relationship between the expected risk of the posterior distribution $Q_t$ and the empirical losses of the entire training process. Actually, research on generalization bounds tailored for online continual learning remains scarce and our bound serves a simple attempt in this area and a extension of Lemma B.1.

## C  Setups and Additional Experiments

### C.1  Experiment Setups

**Memory buffers.** Before delving into the specifics experimental results, it is essential to highlight a significant distinction in the utilization of memory buffers between this paper and other works. Traditional methods typically employ a real-time accessible memory buffer, where at each time step $t$, the model receives a mini-batch of data $X \cup X^b$, drawn i.i.d from $\mathcal{D}_t$ and the memory buffer $\mathcal{B}$, respectively. However, in this work, we impose limitations on the number of requests allowed to retrieve data from the memory buffer. Furthermore, we will assess the time and storage overhead incurred by any additional computations, including data augmentation, knowledge distillation, and gradient calculations. For our experiments, we employ three distinct memory sizes along with their corresponding experience replay frequencies, as presented in each respective table. In contrast to existing replay-based methods that sample a small batch of data from the memory buffer at every training iteration, we evaluate the performance of OCL methods under the assumption that they only have access to the memory buffer every 10, 50, 100 or 500 training iterations. This approach allows us to evaluate the methods under various throughput requirements and is more aligned with the off-site storage of data and models in real-world scenarios.

**Datasets.** We use 6 image classification datasets in the evaluation including CIFAR10, CIFAR100, EuroSat, CLEAR10, CLEAR100 and ImageNet[44, 36, 47, 23]. CIFAR10 has 10 classes with 40,000 for training and 10,000 for testing. It is split into 5 disjoint tasks with 2 classes per task. CIFAR100 has 100 classes with 40,000 for training and 10,000 for testing. It is split into 20 disjoint tasks with 5 classes per task. EuroSat has 10 classes with 17,799 for training and 7,000 for testing. It is split into 5 disjoint tasks with 2 classes per task. CLEAR10, CLEAR100, two continual image classification benchmark datasets with a natural temporal evolution of visual concepts in the real world that spans a decade (2004-2014). We adopt the "streaming" protocols for CL that always test on both seen data and data in the (near) future. ImageNet has 1000 classes with 1,281,167 for training and 50,000 for testing. It is split into 200 disjoint tasks with 5 classes per task. All the methods are trained in a supervised manner and tested on seen classes at any given time. In our experiments, we employ a blurry task boundary as suggested by [43] instead of the conventional disjoint task boundary to better reflect realistic and practical scenarios. Specifically, in the process of data arrival, there is partial overlap (set at 10%) between the data at the boundaries of different tasks, rather than being completely disjoint.

**Implementation details.** For CIFAR10, CIFAR100, and EuroSat, we utilize the ViT-Tiny as a backbone and ViT-Base [26] for CLEAR10, CLEAR100 and ImageNet. We train the model with AdamW optimizer for all the datasets and comparing methods. For all the methods compared, we set the same batch size (10) and replay batch size (10) for fair comparisons. We reproduce all baselines in the same environment with their source code and default settings. For the methods requiring a real-time memory buffer to compute some exclusive variables, we ensure the correct calculation of these variables by increasing the frequency of access to the memory while ensuring a same replay frequency. For the pre-trained models used in Table1, 2, 3,4 and 5, we use the MAE pre-trained initialization [35] for ImageNet and supervised pre-trained initialization for other datasets.

**Compared baselines.** We conducted a comparison of our NCE approach with 13 baselines, as shown in Table5, consisting of 8 replay-based OCL baselines and 5 offline CL baselines. To ensure a fair comparison, we implemented a vanilla experience replay on the 3 offline CL baselines, running all approaches for one epoch.

**Evaluation metrics.** The traditional metric, Average accuracy ($A_{avg}$), is commonly used in continual learning. However, $A_{avg}$ only provides information about the model's performance at specific moments of task transitions, which may occur only 5 to 10 times in most OCL setups. This temporal sparsity of measurement makes it insufficient to deduce conclusions about the model's any-time inference capability. In this paper, we use alternative evaluation metrics: Area Under the Curve of Accuracy ($A_{AUC}$) and Last Accuracy. Inspired by the work of [43], we measure accuracy more frequently by evaluating it after every $\Delta n$ samples, instead of only at discrete task transitions. This new metric is equivalent to the area under the curve (AUC) of the accuracy-to-# of samples curve for continual learning methods, with $\Delta n = 1$. We refer to it as Area under the curve of accuracy ($A_{AUC}$), calculated as $A_{AUC} = \sum_{i=1}^{t} f(i \cdot \Delta n) \cdot \Delta n$. Additionally, we include Last Accuracy as another evaluation metric. Last Accuracy simply refers to the model's accuracy after it has processed all the data in the data streams.

**Device.** All the experiments are implemented on NVIDIA RTX2080ti and RTX4090ti. It is notable that all results on training efficiency, model throughput and inference time are done on RTX2080ti.

## C.2 More Detailed Experimental Results

### C.2.1 Last Accuracy

In addition to $A_{AUC}$, we also evaluate the last accuracy of various OCL methods. We include these two evaluations because $A_{AUC}$ allows us to assess the real-time performance of the model, while the last accuracy measurement reflects the model's performance after processing the entire data stream. As shown in Table4 and 5, Even without employing data augmentation and knowledge distillation, our NsCE framework still achieves comparable results. This is particularly evident when faced with more stringent constraints on memory buffer size and replay frequency.

### C.2.2 NsCE Lite.

In addition to enhancing model throughput through constraining memory replay, we also consider the possibility of not fine-tuning the entire network since we have already utilized a pre-trained model. However, when faced with large-scale data streams with changing data distributions, it becomes challenging for the model to adapt to new data without fine-tuning. In such cases, we evaluate the

Table 4: *Last Accuracy* on synthetic online class-incremental setting. Best is highlighted in **bold**, second best is shown underlined.

| Method | CIFAR-10 | | | CIFAR-100 | | | EuroSat | | |
|---|---|---|---|---|---|---|---|---|---|
| | $M = 0.1k$ $Freq = 1/100$ | $M = 0.2k$ $Freq = 1/50$ | $M = 0.5k$ $Freq = 1/10$ | $M = 0.5k$ $Freq = 1/100$ | $M = 1k$ $Freq = 1/50$ | $M = 2k$ $Freq = 1/10$ | $M = 0.1k$ $Freq = 1/100$ | $M = 0.2k$ $Freq = 1/50$ | $M = 0.5k$ $Freq = 1/10$ |
| iCaRL[58] | 90.1±0.2 | 90.0±0.1 | 92.3±0.3 | 69.6±0.4 | 70.2±0.7 | 73.1±0.2 | 67.7±0.8 | 77.9±1.0 | 87.5±0.4 |
| EWC[41] | 85.5±0.4 | 92.4±0.8 | 94.2±1.0 | 67.9±0.8 | 66.0±1.1 | 70.4±1.3 | 75.7±1.1 | 84.5±0.9 | 89.2±1.0 |
| DER++[10] | 87.7±1.4 | 91.1±0.9 | 93.0±1.1 | 67.8±1.7 | 71.5±1.0 | 73.7±0.9 | 66.9±4.3 | 84.7±1.4 | 87.4±2.0 |
| PASS[82] | 91.2±1.1 | 92.0±0.9 | 94.4±0.7 | 69.0±1.4 | 71.7±1.2 | 72.0±0.9 | 70.9±0.8 | 84.5±1.0 | 88.7±1.0 |
| MC-SGD w/ SAM[51] | 90.7±0.6 | 91.5±0.7 | 94.2±0.4 | 70.0±0.8 | 73.1±0.8 | 74.0±1.6 | 75.3±0.4 | 88.9±0.7 | 94.0±1.5 |
| AGEM[15] | 84.3±0.3 | 90.4±0.2 | 93.7±0.9 | 68.2±0.4 | 67.8±0.3 | 73.5±0.1 | 75.6±1.1 | 87.6±0.8 | 91.8±0.7 |
| ER[16] | 91.3±0.7 | 92.0±0.4 | **94.9±0.2** | 73.5±0.6 | 73.3±0.5 | 73.6±0.4 | 76.3±0.8 | 89.8±1.0 | 93.4±0.9 |
| MIR[4] | 92.0±1.0 | 92.1±0.8 | 94.1±1.1 | 68.4±0.8 | **74.1±0.8** | **74.7±0.8** | 74.4±2.4 | 88.4±1.6 | 90.0±0.8 |
| ASER[60] | 86.2±0.9 | 90.2±1.2 | 93.0±1.1 | 67.9±0.8 | 71.0±0.4 | 72.3±0.8 | 74.3±0.8 | 85.9±0.5 | 92.9±0.8 |
| SCR[49] | 89.9±0.6 | 92.2±0.5 | 93.5±1.0 | 69.9±0.9 | 71.7±0.5 | 73.1±0.3 | 75.8±0.8 | 87.8±0.7 | 94.2±1.1 |
| DVC w/o Aug[29] | 90.1±0.8 | 91.9±0.9 | 92.7±0.8 | 71.4±0.2 | 71.0±0.6 | 73.5±1.0 | 74.2±4.2 | 85.7±1.0 | 92.3±0.8 |
| DVC w/ Aug[29] | 90.6±0.9 | 91.3±0.7 | 94.1±0.4 | 72.7±0.6 | 72.8±0.8 | 73.4±0.6 | 75.1±0.4 | 88.9±0.8 | 92.4±1.2 |
| OCM w/o Aug[31] | 84.1±1.3 | 83.3±1.4 | 92.0±1.1 | 64.2±0.8 | 55.0±1.6 | 64.7±0.4 | 72.8±1.2 | 78.7±1.0 | 80.4±0.6 |
| OCM w/ Aug[31] | 85.6±1.1 | 89.5±1.4 | 93.6±0.5 | **73.7±0.8** | 73.5±1.0 | 73.9±0.5 | 74.1±1.0 | 86.4±1.5 | 93.5±0.7 |
| OnPro w/Aug[68] | 92.2±0.9 | 92.1±0.6 | 94.1±0.5 | 69.4±0.5 | 73.7±0.8 | 74.1±0.6 | 76.8±2.4 | 88.6±0.7 | 93.3±1.1 |
| NsCE | **93.1±0.5** | **93.0±1.4** | 93.1±1.2 | 70.8±1.5 | 73.9±1.7 | 73.7±1.4 | **84.9±1.7** | **91.7±0.8** | **94.4±0.6** |

Table 5: *Last Accuracy* on real-world online domain-incremental setting and large scale data stream. Best is highlighted in **bold**, second best is shown underlined.

| Method | CLEAR-10 | | CLEAR-100 | | ImageNet |
|---|---|---|---|---|---|
| | $M = 0.1k$ $Freq = 1/100$ | $M = 0.2k$ $Freq = 1/50$ | $M = 1k$ $Freq = 1/100$ | $M = 2k$ $Freq = 1/50$ | $M = 10k$ $Freq = 1/500$ |
| ER | 93.4±0.2 | 93.5±0.6 | 88.5±0.9 | 88.1±0.2 | 47.0±0.6 |
| DER | 92.7±0.4 | 93.4±0.8 | 87.9±0.4 | 88.9±0.3 | 43.8±0.7 |
| EWC | 93.0±0.6 | 92.1±0.7 | 89.3±1.4 | **89.6±0.9** | 46.0±0.4 |
| iCarl | 91.4±0.9 | 92.8±1.2 | 84.9±1.3 | 85.9±0.8 | 47.9±1.0 |
| SCR | 89.4±0.6 | 89.8±0.6 | 83.4±0.6 | 87.0±1.1 | 46.3±1.4 |
| OCM | 92.1±0.5 | 92.7±1.3 | 87.9±0.9 | 86.7±0.6 | ××××× |
| DVC | 91.3±0.8 | 91.9±0.4 | 88.0±0.6 | 88.1±0.5 | 45.9±0.5 |
| OnPro | 92.2±1.2 | 93.5±1.6 | 88.0±0.9 | 88.3±0.7 | 47.1±0.6 |
| NsCE | **93.9±0.7** | **94.2±1.1** | **90.1±0.8** | 89.2±1.6 | **49.8±2.4** |

features learned by our model (Eq.2) during training on the data stream. If the model has acquired sufficiently discriminative features for the current task, we believe that only updating the classifier layer would suffice to achieve optimal results. We refer to this lightweight framework as NsCE Lite, detailed in Algorithm1.

We conducted tests on our lightweight version of NsCE using the smallest memory buffer and lowest replay frequency on six datasets, as presented in Table6. In most cases, this lightweight framework also achieves comparable results with NsCE, particularly on relatively simple datasets like CIFAR10 and EuroSat, where NsCE Lite even outperforms the original NsCE approach. The potential reason for this improvement may be that when NsCE Lite encounters simpler datasets, despite constraining the sparseness of the classifier parameters with the dataset, model's myopia caused by intensified parameter sparsity exists not only in the classifier but also in the feature extraction process. This becomes especially apparent when the model acquires highly separable features. Further training may cause the model to excessively focus on discriminative features that lack generalization ability. Hence, introducing a detach operation to the feature extractor $f(\cdot)$ can effectively mitigate the model's myopia, especially when the model has attained satisfactory performance on the current task. By detaching the feature extractor, the model can retain the learned features while allowing for independent updates and adjustments to the classification layer or other components.

## C.3 Discussions on Utilization of Pre-trained Models

**Model's ignorance and myopia: new perspectives.** The current performance bottleneck of the OCL method serves as the initial motivation for our study on the application of pre-trained models in OCL. As shown by results in Table7 from [68], even the best methods can only reach about 30% on CIFAR100 and 20% accuracy on TinyImageNet. Although the exploration of these methods may be meaningful to the community, such performance is completely unworthy of discussion for

Table 6: Comparison results between our proposed NsCE and NsCE Lite ($A_{AUC}$). The methods that exhibit the best performance with pre-trained models are highlighted in **bold**.

| Model | CIFAR10 $M = 0.1k, Freq = 1/100$ | CIFAR100 $M = 0.5k, Freq = 1/100$ | EuroSat $M = 0.1k, Freq = 1/100$ | CLEAR10 $M = 0.1k, Freq = 1/100$ | CLEAR100 $M = 1k, Freq = 1/100$ | ImageNet $M = 10k, Freq = 1/500$ |
|---|---|---|---|---|---|---|
| NsCE | 89.9±0.4 | **74.1±0.7** | 75.7±0.4 | **89.2±0.7** | **84.3±0.4** | **61.6±0.7** |
| NsCE Lite | **90.7±0.6** | 72.9±0.9 | **76.1±0.5** | 88.0±0.6 | 82.9±1.2 | 60.9±0.8 |

**Algorithm 1** NsCE (Lite)

---
**Input:** Data stream $\mathfrak{D}$, encoder $\theta_f$, classifier $\phi$
**Initialization:** Memory buffer $\mathcal{M} \leftarrow \{\}, acc_t = 0$
**for** $t = 1$ **to** $T$ **do**
    **for** each mini-batch $X$ in $D_t$ **do**
        $M \leftarrow$ Update$(M, X)$
        **if** $acc_t > threshold$ **then**
            $\theta_f$.detach()
        **end if**
        $p = \phi(\theta_f(X))$, $z = \theta_f(X)$
        Compute online class mean $\mu$ and $y^*$ by Eq.2
        $\theta_f, \theta_\phi \leftarrow \mathcal{L}_{ce} + \gamma(\mathcal{L}_p + \mathcal{L}_s)$ by Eq.7
        **if** Replay **then**
            Compute Confusion Matrix on Memory buffer
            $\theta_f, \theta_\phi \leftarrow \mathcal{L}_b$ by Eq.8
        **end if**
        Caculate the accuracy $acc_t$ on current task by $y*$
    **end for**
**end for**

---

practical problems. Furthermore, when we review previous work from the perspective of **model's ignorance and myopia**, we observe that many techniques originally developed for continuous learning, such as knowledge distillation and dark experience replay, may not be fully applicable to OCL scenarios. In OCL, the model requires more than just relying on past cognition. It needs the flexibility to dynamically adjust and update its features and classification criteria. The improvement in performance of OCL methods often stems from alleviating model ignorance through replaying past data and incorporating augmentation methods. These approaches help the model adapt and refine its representations, reducing the impact of myopia and enabling better performance in OCL settings.

**Inspiration of using pre-trained models.** Inspired by how humans learn quickly and effectively, we realize our ability to recognize new class is typically built upon fundamental cognitive abilities and prior knowledge. In fact, for most intelligent life forms, the process of learning begins with the acquisition of fundamental concepts and knowledge. For instance, humans possess innate abilities for perception and an instinctual drive to seek advantages while avoiding disadvantages. These foundational aspects of learning form the basis upon which more complex cognitive abilities and knowledge are built. We anticipate that pre-trained models, which have demonstrated success across various domains, can play a similar role as fundamental knowledge that enables a high-throughput, high-performance OCL model supporting any-time inference.

Table 7: Average accuracy of state-of-the-art methods without the pre-trained initialization. (The results are directly copied from [68])

| Method | CIFAR-10 | | | CIFAR-100 | | | TinyImageNet | | |
|---|---|---|---|---|---|---|---|---|---|
| | $M = 0.1k$ | $M = 0.2k$ | $M = 0.5k$ | $M = 0.5k$ | $M = 1k$ | $M = 2k$ | $M = 1k$ | $M = 2k$ | $M = 4k$ |
| iCaRL[58] | 31.0±1.2 | 33.9±0.9 | 42.0±0.9 | 12.8±0.4 | 16.5±0.4 | 17.6±0.5 | 5.0±0.3 | 6.6±0.4 | 7.8±0.4 |
| DER++[10] | 31.5±2.9 | 39.7±2.7 | 50.9±1.8 | 16.0±0.6 | 21.4±0.9 | 23.9±1.0 | 3.7±0.4 | 5.1±0.8 | 6.8±0.6 |
| PASS[82] | 33.7±2.2 | 33.7±2.2 | 33.7±2.2 | 7.5±0.7 | 7.5±0.7 | 7.5±0.7 | 0.5±0.1 | 0.5±0.1 | 0.5±0.1 |
| AGEM[15] | 17.7±0.3 | 17.5±0.3 | 17.5±0.2 | 5.8±0.1 | 5.9±0.1 | 5.8±0.1 | 0.8±0.1 | 0.8±0.1 | 0.8±0.1 |
| GSS[6] | 18.4±0.2 | 19.4±0.7 | 25.2±0.9 | 8.1±0.2 | 9.4±0.5 | 10.1±0.8 | 1.1±0.1 | 1.5±0.1 | 2.4±0.4 |
| ER[16] | 19.4±0.6 | 20.9±0.9 | 26.0±1.2 | 8.7±0.3 | 9.9±0.5 | 10.7±0.8 | 1.2±0.1 | 1.5±0.2 | 2.0±0.2 |
| MIR[4] | 20.7±0.7 | 23.5±0.8 | 29.9±1.2 | 9.7±0.3 | 11.2±0.4 | 13.0±0.7 | 1.4±0.1 | 1.9±0.2 | 2.9±0.3 |
| GDumb[56] | 23.3±1.3 | 27.1±0.7 | 34.0±0.8 | 8.2±0.2 | 11.0±0.4 | 15.3±0.3 | 4.6±0.3 | 6.6±0.2 | 10.0±0.3 |
| ASER[60] | 20.0±1.0 | 22.8±0.6 | 31.6±1.1 | 11.0±0.3 | 13.5±0.3 | 17.6±0.4 | 2.2±0.1 | 4.2±0.6 | 8.4±0.7 |
| SCR[49] | 40.2±1.3 | 48.5±1.5 | 59.1±1.3 | 19.3±0.6 | 26.5±0.5 | 32.7±0.3 | 8.9±0.3 | 14.7±0.3 | 19.5±0.3 |
| CoPE[20] | 33.5±3.2 | 37.3±2.2 | 42.9±3.5 | 11.6±0.7 | 14.6±1.3 | 16.8±0.9 | 2.1±0.3 | 2.3±0.4 | 2.5±0.3 |
| DVC[29] | 35.2±1.7 | 41.6±2.7 | 53.8±2.2 | 15.4±0.7 | 20.3±1.0 | 25.2±1.6 | 4.9±0.6 | 7.5±0.5 | 10.9±1.1 |
| OCM[31] | 47.5±1.7 | 59.6±0.4 | 70.1±1.5 | 19.7±0.5 | 27.4±0.3 | 34.4±0.5 | 10.8±0.4 | 15.4±0.4 | 20.9±0.7 |
| OnPro[68] | 57.8±1.1 | 65.5±1.0 | 72.6±0.8 | 22.7±0.7 | 30.0±0.4 | 35.9±0.6 | 11.9±0.3 | 16.9±0.4 | 22.1±0.4 |

**Pre-trained model is not a one-size-fits-all solution.** Although pre-trained models have demonstrated significant performance improvements across various datasets compared with the performance trained from scratch (as illustrated in Table 7), it is crucial to acknowledge that they are not a one-size-fits-all solution. The limitations manifest in multiple aspects, as depicted in Table 8, 9 and

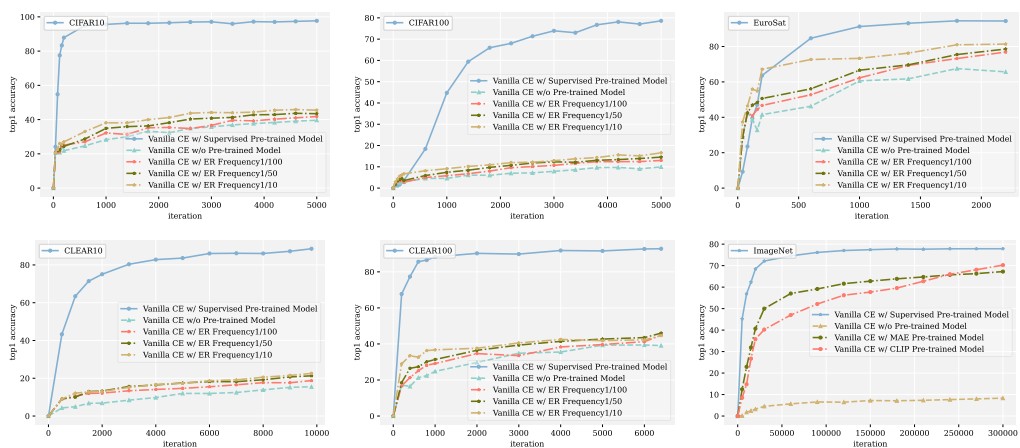

Figure 7: We evaluate the real-time accuracy of models on currently seen classes (w/) and (w/o) pre-trained models under our designed **single task setting**, as well as the impact of experience replay frequency on CIFAR, EuroSAT, CLEAR and ImageNet.

Table 8: Performance ($A_{AUC}$) under our **single task setting**. We illustrate the impact brought by pre-trained models (models pre-trained on ImageNet by Masked Auto Encoder[35]) with different network architectures over various datasets. The networks that exhibit the best performance with pre-trained models are highlighted in **bold**, while the networks that achieve the best performance without pre-trained models are shown underlined.

| Model | CIFAR10 | CIFAR100 | EuroSat | SVHN | TissueMNIST |
|---|---|---|---|---|---|
| ViT-T w/o pretrain | 31.31 | 9.77 | 55.71 | 54.16 | 43.68 |
| ViT-T w/ pretrain | 93.54 | 61.97 | 76.09 | 88.24 | 59.84 |
| Δ | **+62.23** | **+52.20** | +20.38 | +34.08 | +16.16 |
| ViT-S w/o pretrain | 37.64 | 6.95 | 51.81 | 36.86 | 42.78 |
| ViT-S w/ pretrain | **90.38** | **79.49** | **78.02** | **93.33** | **60.04** |
| Δ | +52.74 | +72.54 | +26.21 | +56.47 | +17.26 |

10, the benefits (or drawbacks) of utilizing a pre-trained model vary depending on the dataset and network architecture employed. Pre-trained models do not consistently yield substantial gains across all datasets. The effectiveness of pre-training depends on several factors, such as the degree of domain similarity between the pre-training and target tasks, the size and quality of the pre-training dataset, some specific characteristics of the target dataset and even the structure of backbone networks matters. For CIFAR, EuroSat, SVHN[53], and TissueMNIST[71], the distributional discrepancy between the pre-training and downstream task data gradually increases. It is evident that pre-trained models tend to provide more benefits when the pre-trained data share common semantics with the downstream tasks. Surprisingly, even for the same dataset and pre-training approach, the choice of model architecture can significantly impact the performance of the model, as observed in Table 8,

Table 9: Performance ($A_{AUC}$) under our **single task setting**. We illustrate the impact brought by pre-trained models (models pre-trained on CLIP[57]) with different network architectures over various datasets. The networks that exhibit the best performance with pre-trained models are highlighted in **bold**, while the networks that achieve the best performance without pre-trained models are shown underlined.

| Model | CIFAR10 | CIFAR100 | EuroSat | SVHN | TissueMNIST |
|---|---|---|---|---|---|
| Res50 w/o pretrain | 38.58 | 14.04 | 37.64 | 88.04 | 43.72 |
| Res50 w/ pretrain | **86.44** | **79.27** | **65.31** | **92.10** | **60.31** |
| Δ | +47.86 | +65.23 | **+27.67** | +4.06 | **+16.59** |
| ViT-S w/o pretrain | 37.64 | 6.95 | 51.81 | 36.86 | 42.78 |
| ViT-S w/ pretrain | 83.70 | 76.87 | 59.36 | 90.09 | 49.37 |
| Δ | **+55.24** | **+69.92** | +7.55 | **+53.23** | +6.59 |

Table 10: Performance ($A_{AUC}$) under our **single task setting**. We illustrate the impact brought by pre-trained models (supervised models pre-trained on ImageNet) with different network architectures over various datasets. The networks that exhibit the best performance with pre-trained models are highlighted in **bold**, while the networks that achieve the best performance without pre-trained models are shown underlined.

| Model | CIFAR10 | CIFAR100 | EuroSat | SVHN | TissueMNIST |
|---|---|---|---|---|---|
| Res18 w/o pretrain | 30.62 | 16.21 | 30.14 | 81.45 | 39.57 |
| Res18 w/ pretrain | 43.60 | 53.41 | 35.68 | 85.60 | 39.40 |
| Δ | +12.98 | +37.20 | +5.54 | +4.15 | -0.17 |
| Res50 w/o pretrain | 38.58 | 14.04 | 37.64 | 88.04 | 43.72 |
| Res50 w/ pretrain | 49.21 | 56.73 | 46.20 | 88.91 | 48.60 |
| Δ | +14.63 | +42.69 | +8.56 | +0.87 | +4.88 |
| WRN28-2 w/o pretrain | 32.10 | 11.17 | 29.52 | 87.69 | 42.38 |
| WRN28-2 w/ pretrain | 46.83 | 37.57 | 36.67 | 87.78 | 39.41 |
| Δ | +14.73 | +26.40 | +7.15 | +0.09 | -2.97 |
| WRN28-8 w/o pretrain | 26.66 | 13.22 | 21.24 | 90.17 | 44.35 |
| WRN28-8 w/ pretrain | 57.13 | 44.23 | 29.91 | 90.93 | 44.05 |
| Δ | +30.47 | +31.01 | +8.67 | +0.76 | -0.30 |
| ViT-T w/o pretrain | 31.31 | 9.77 | 55.71 | 54.16 | 43.68 |
| ViT-T w/ pretrain | 91.49 | 57.55 | 76.40 | 90.07 | 53.35 |
| Δ | **+60.18** | +47.78 | +10.70 | +35.91 | +9.67 |
| ViT-S w/o pretrain | 37.64 | 6.95 | 51.81 | 36.86 | 42.78 |
| ViT-S w/ pretrain | **92.88** | **76.87** | **79.40** | **95.11** | **57.64** |
| Δ | +55.24 | **+69.92** | **+17.59** | **+58.25** | **+14.86** |

9 and 10 on SVHN and TissueMNIST. These findings highlight the nuanced nature of leveraging pre-trained models. When comparing networks based on convolutional neural networks (CNN), it is evident that transformer-based models tend to derive more benefits from pre-trained models. Furthermore, we also discover that different pre-training methods also exert a significant influence on the model's performance. However, despite conducting these experiments, we have not yet been able to discern definitive rules for selecting pre-trained models. Careful consideration and experimentation are necessary to identify the optimal combination of pre-training and downstream task settings for achieving the desired performance improvements. The overall efficacy of pre-training is influenced by various elements, including the domain alignment between the pre-training and target tasks, the volume and integrity of the pre-training dataset, specific pre-training scheme, particular attributes of the target dataset and the architecture of the backbone network. Generally, we posit that possessing insight into the expected data distribution of upcoming tasks enables the selection of a pre-trained model trained on a comparable distribution, which is a prudent and dependable approach. We believe the selection of appropriate pre-trained models remains an important open question for many areas including OCL.

## D   Detailed Discussions on Efficiency and Feasibility of Current OCL Methods

In this section, we present empirical observations on the efficiency and feasibility of current OCL methods. We will discuss these observations from several key aspects: requirements on memory buffer, model throughput, and performance. By examining these aspects, we aim to provide insights into the practicality and effectiveness of existing OCL methods in real-world applications.

### D.1   Requirements on Real-time Memory Buffers

As we stated before, OCL serves as a more realistic extension of continual learning, unlike traditional batch learning, where the entire dataset for each task is available upfront, it operates in scenarios where data distributions dynamically change over time. However, we find that, there is very limited research that specifically addresses the accessibility of memory buffers during training in the context of OCL. In this study, we argue that such an assumption is highly unrealistic in a real-world environment. Most existing replay-based OCL methods exhibit some flaws when applied to real-world applications:

(1) As illustrated in Figure8, a notable characteristic of replay-based methods is their tendency to sample a larger proportion of data from the memory buffer compared to the incoming data stream. However, such a continuous sampling process significantly restricts the throughput of the model for streaming data. It has been observed that the time required to retrain memory data is typically 3-5 times longer than that for new data. Similarly, some data augmentations also significantly reduce

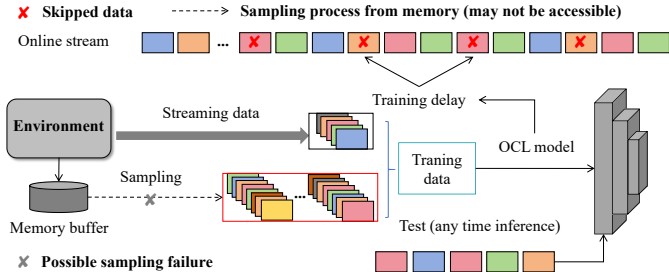

Figure 8: Common framework of replay-based OCL methods. Possible sampling failure and training delay due to the mismatch between model training speed and the data stream flow rate are two primary concerns.

the model throughput. This prolonged training time not only results in increased training delays but also leads to more skipped data, reducing the amount of available data that can be effectively utilized within a given time frame, as highlighted in[28, 80].

(2) Furthermore, there is a typical assumption that the memory buffer needs to store dozens of samples for each class to help the model to enable efficient review of past classes. However, when dealing with large-scale datasets such as ImageNet [23], the storage overhead becomes impractical, particularly for data that needs to be real-time accessible and stored in local memory. Not to mention that in real-world scenarios, the number of categories in the data stream we encounter is constantly increasing. Such limitations become evident when the system lacks continuous access to past data, severely restricting the model's learning capacity, as illustrated in Figure 7. Additionally, most existing methods actually require the memory buffer, model and data being processed to be stored in GPU memory simultaneously to avoid latency during access. This discrepancy between the existing methods and the practical scenario further diminish the practicality of current OCL methods.

(3) In real-world applications, such as autonomous vehicles [38] or sensor networks [42], ensuring the real-time accessibility of the memory buffer presents a significant challenge, especially when the learning system is deployed in terminal equipment. The constraints of computing resources, privacy concerns and network connectivity all hinder the sampling process in the memory buffer, thereby diminishing the feasibility of existing OCL methods. For example, the latency caused by data transfer makes it difficult for the model to synchronize and obtain data from both the memory buffer and the incoming data stream. Additionally, due to privacy and copyright concerns, in most cases, we cannot store data from the data stream arbitrarily. For instance, in real-world autonomous driving scenarios, data cannot be uploaded to data centers at any time. Meanwhile, the data stored in data centers usually undergoes strict scrutiny to avoid the risk of infringing on privacy or the commercial copyright of another party.

In our research, we simulate the situation where the memory buffer, model and data stream are stored separately by limiting the number of accesses to the memory, which means we cannot replay the desired data at any given moment. Nevertheless, it is important to acknowledge that there may still be a gap between the scenarios we simulated and real-life applications. However, we firmly believe that taking this step is beneficial to the community.

## D.2    Model Throughput and Performance

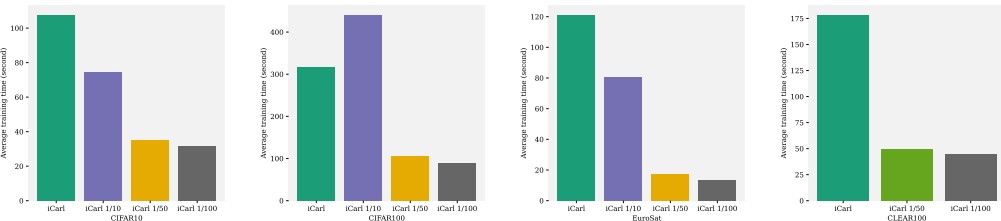

Figure 9: Training time of iCarl[58] under different replay frequencies across datasets.

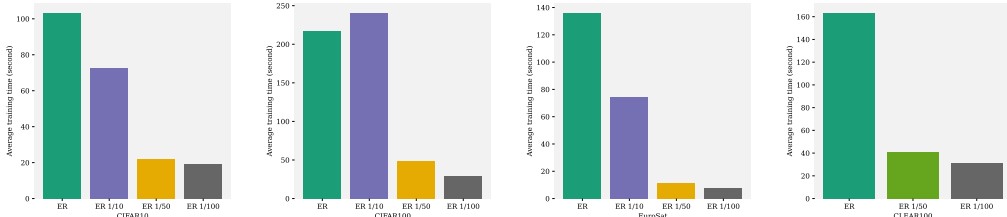

Figure 10: Training time of Experience Replay (ER)[16] under different replay frequencies across datasets.

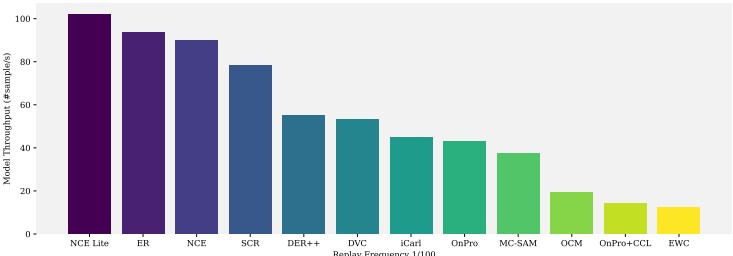

Figure 11: Averaged model throughput of 11 OCL methods on 6 datasets (Replay frequency as 1/100).

In recent years, various OCL methods have been proposed [48, 31, 68, 66, 79], among them, replay-based techniques that interleave past experiences with new data have emerged as a predominant component. It aims to consolidate feature learning from earlier tasks while mitigating catastrophic forgetting through the constant re-exposure of few old data. However, the evaluation of OCL methods often overlooks assessment of model throughput, a critical metric especially for data streams with different coming speed. To assess the performance and efficiency of popular OCL methods more effectively, we first record the running time as shown in Figure 9 and Figure 10. It is evident that as the replay frequency increases, the training time of the model (every 200 training iterations) also significantly increases. Consequently, this leads to a substantial reduction in the model's throughput. In comparison, our experimental settings, including frequencies of $1/50$ and $1/100$, considerably shorten the training time and enhance the model's throughput compared to fetching data from the memory buffer for each training iteration. Moreover, we evaluate the averaged model throughput of 11 OCL methods on 6 datasets. As shown in Figure 11, our proposed NsCE and NsCE Lite improve the model throughput while ensuring good performance.

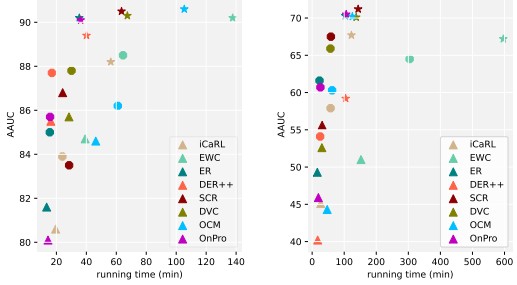

Figure 12: $A_{AUC}$ (Area Under the Curve of Accuracy) and running time on CIFAR10 and CIFAR100. ▲,●,★ represents for different replay frequency of $1/100, 1/50, 1/10$.

We also visualize the running time and $A_{AUC}$ of various OCL models with a pre-trained initialization. As shown in Figure 12, existing methods indeed achieved improvements in model performance, but

they often come at the cost of slower training speed. Interestingly, as shown in prior work[28], if we utilize the available extra time to increase the frequency of replay and train the model multiple times on the memory, the performance gains are often greater compared to using state-of-the-art techniques such as various regularization or knowledge distillation techniques.

In addition to training time, we also compare the inference time between our NsCE method and existing approaches as it also serves as an important metric when doing the real-time inference. As indicated in Table11, our method achieves an inference time comparable to ER. However, due to the requirement to compute feature similarity or the need for extra projector, methods like iCaRL and OnPro exhibit slower inference speeds.

Table 11: Comparison on the inference time between our NsCE and some popular methods.

| Methods | CIFAR10 | EuroSat | ImageNet |
|---------|---------|---------|----------|
| ER | 22s | 16s | 6min43s |
| iCaRL | 31s | 26s | 9min45s |
| OnPro | 24s | 20s | 7min09s |
| NCE | 22s | 18s | 6min50s |

### D.3 Conclusion

In addition to efficiency, achieving a balance between model performance, throughput, and practicality is of great concern. The aforementioned practical limitations highlight the necessity for innovative approaches that can adapt to resource-constrained environments and effectively address the challenges in OCL. Considering the difficulties encountered in real-world scenarios, a simple alternative is to limit the frequency of memory access throughout the entire training process. This approach not only improves training throughput but also eliminates the need for real-time storage, thereby alleviating requirements related to hardware specifications, network connectivity, and privacy concerns.

However, this alternative approach introduces new challenges in avoiding model myopia and potential forgetting. While pre-training models can expedite the learning of valuable features, sampling less from memory makes the model to excessively concentrate on the current task, increasing the risk of model myopia and catastrophic forgetting.

## E  Forgetting Phenomenon

We meticulously record the changes in model classification results when using a linear classifier without pre-training initialization, using a linear classifier with pre-training initialization, and using a prototype classifier with pre-training initialization. As demonstrated by Figure14 and Figure15, pre-trained initialization allows the model to retain previously learned discriminant information without indiscriminately dividing the data into the current class. In our evaluation, we specifically assess the classifier results of CIFAR10 using linear softmax and the NCM prototype classifier. To gain insights into the model's classification performance, we visualize the classification results on the test set at the beginning and end of each task, as shown in Figure14. We calculate the model's classification confusion matrix to analyze the results.Our observations reveal the following:

- Pre-trained initialization helps the model rapidly achieve performance on the current task while providing a broader perspective to avoid mindlessly classifying past classes as part of the current task (as the comparison in red and blue in Figure15).

- The linear softmax classifier demonstrates a quicker acquisition of improved discriminative abilities for data within the current task compared to the NCM classifier (as the comparison in green and blue in Figure15). However, it is more prone to misclassifying categories from previous tasks as belonging to the current task, resulting in a decline in overall performance.

- When using a pre-trained model and having continuous access to the memory buffer, as illustrated in Figure14, we can quickly achieve good overall performance.

To further clarify the difference of our recognized **model's myopia** and the catastrophic forgetting, we implement existing anti-forgetting techniques against myopia and benchmark our NsCE framework against prevalent methods both with and without experience replay. Our results in Table12 show that popular gradient-based regularization methods such as EWC and AGEM do not effectively prevent performance degradation. Their performance are only on par with a simple supervised

Table 12: Comparison of anti-forgetting techniques and our method.

| Methods | CIFAR10 w/ ER | CIFAR10 w/o ER | CIFAR100 w/ ER | CIFAR100 w/o ER | EuroSat w/ ER | EuroSat w/o ER |
|---|---|---|---|---|---|---|
| Baseline | 82.6 | 71.6 | 61.3 | 39.3 | 58.6 | 38.8 |
| EWC | 81.7 | 70.9 | 60.7 | 40.4 | 61.0 | 33.4 |
| AGEM | 78.6 | 72.3 | 50.2 | 37.9 | 56.4 | 39.7 |
| SCR | 83.8 | 70.2 | 61.5 | 40.4 | 52.1 | 40.4 |
| OnPro | 81.1 | 71.4 | 62.9 | 42.0 | 52.8 | 41.8 |
| **NsCE** | **86.2** | **79.8** | **66.1** | **46.8** | **72.4** | **45.6** |

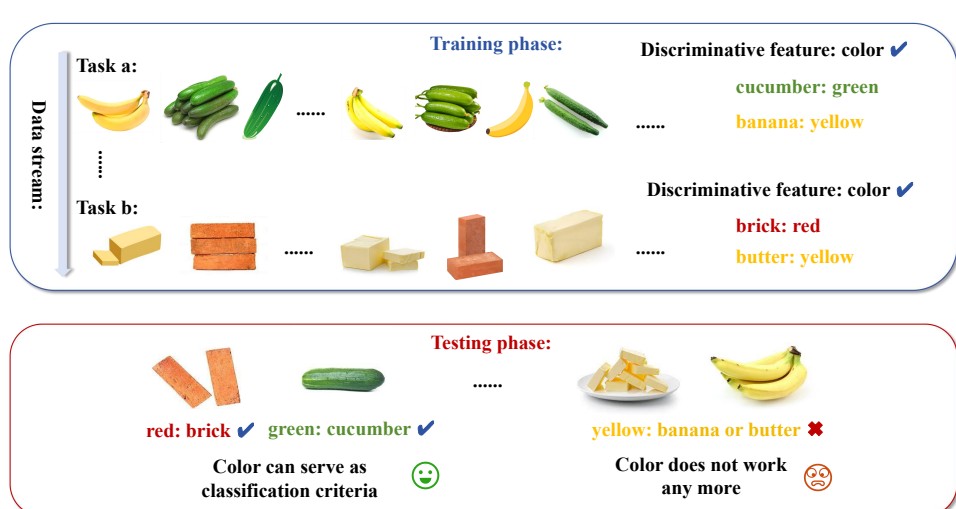

Figure 13: Color, which is the most discriminative feature for task a (banana vs. cucumber) and task b (butter vs. brick), is precisely the reason why the model confuses butter and banana.

learning baseline and it's the same case with or without pre-trained initialization. Additionally, techniques like SCR and OnPro, which use contrastive learning to improve feature discrimination, do not consistently enhance performance. Notably, the gains seen with SCR and OnPro largely stem from use of augmented samples, a tactic not used in earlier gradient-based methods or our NsCE approach. These findings underscore the idea that **model myopia** is a more pressing concern than the often-addressed catastrophic forgetting in the context of OCL.

In addition to the empirical evidence, we give a simplified example of the **model's myopia**. As shown in Figure 13, the discriminative features or attributes for the current task may exactly be the cause of confusion when dealing with future categories, which means the model's cognition for each class must dynamically evolve with the arrival of new data.

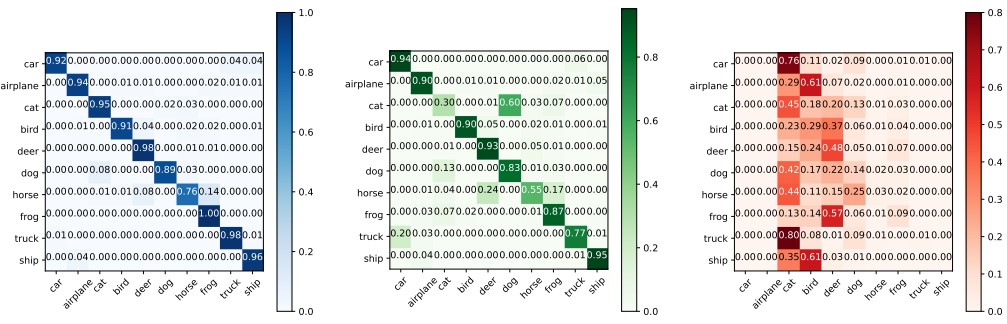

Figure 14: Normalized confusion matrix after fine-tuning on a memory buffer with a size of 500.

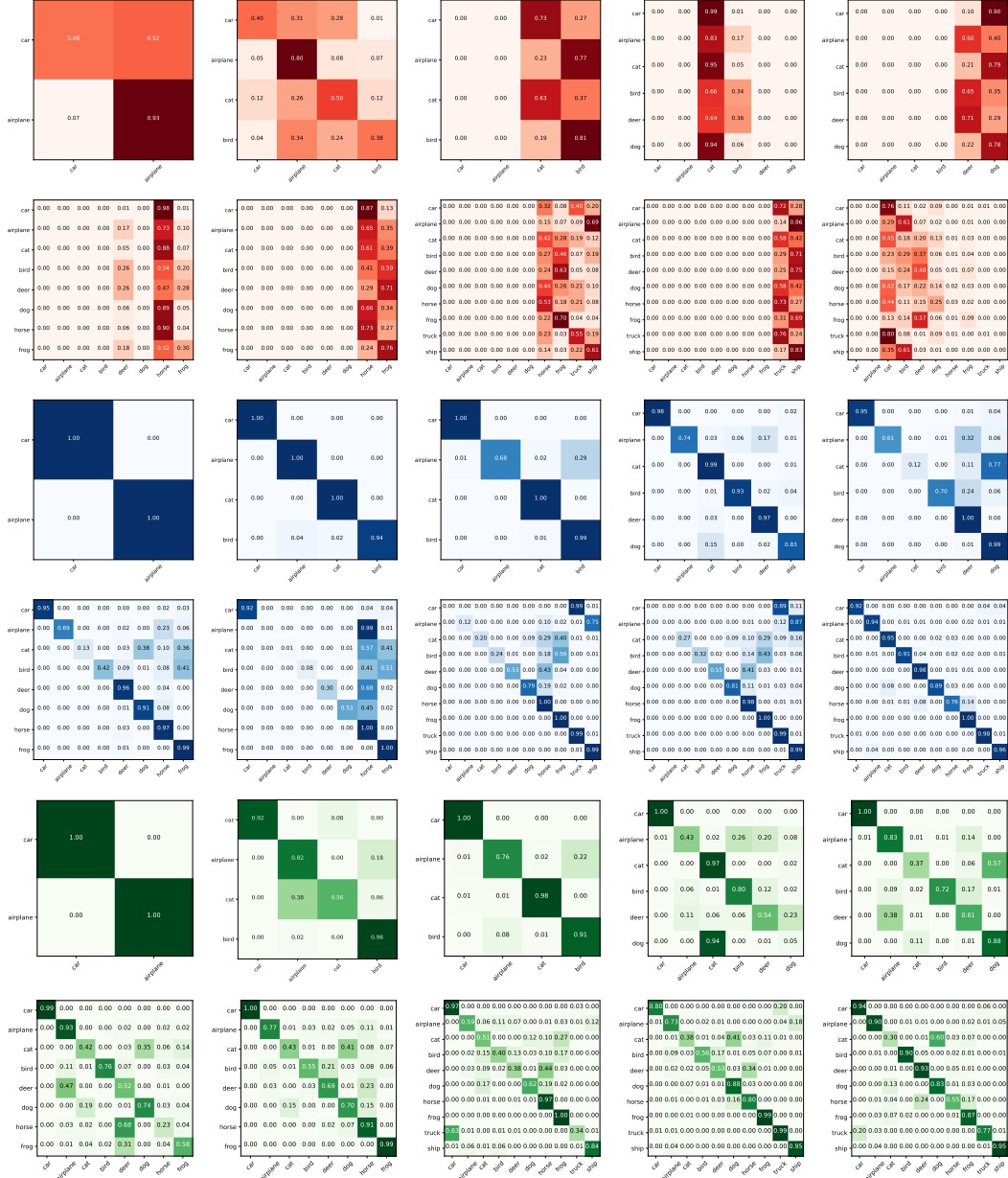

Figure 15: The normalized confusion matrix (CIFAR10) evolution of linear softmax classifier without pre-trained initialization (red) and NCM classifier (green), linear softmax classifier (blue) with supervised pre-trained models on ImageNet.

