# OpenReview forum: "Forgetting, Ignorance or Myopia: Revisiting Key Challenges in Online Continual Learning"
_NeurIPS.cc/2024/Conference — NeurIPS 2024 poster_

### Official Review · Reviewer_abk8 · 2024-07-07

**Soundness:** 3
**Presentation:** 3
**Contribution:** 3
**Rating:** 6
**Confidence:** 5

**Summary:**

This paper revisits the core challenges of Online Continual Learning (OCL) in high-speed data stream environments, identifying two significant obstacles: the model's ignorance and myopia. It then introduces a non-sparse classifier evolution framework (NsCE) designed to effectively address these issues. Additionally, the authors offer some theoretical guarantees from a PAC-Bayes perspective, providing insights into the robustness and generalization capabilities of the proposed method.

**Strengths:**

1. The analysis of factors beyond forgetting in OCL is enlightening, especially in highlighting the suboptimal performance of current approaches.
2. The inclusion of model throughput as a factor in OCL is crucial, and the analysis from a Pac-Bayes perspective offers compelling insights.
3. The examination of sparsity and the proposed Non-sparse Classifier Evolution (NsCE) in continual learning is noteworthy. The method is straightforward, easy to implement, and has proven effective in experiments.
4. The paper is generally well-structured and easy to understand.

**Weaknesses:**

1. While the authors have imposed constraints on the number and frequency of memory buffer accesses, these conditions still mimic laboratory settings rather than real-world applications. It would be beneficial to provide examples of real-world scenarios where such restrictions are applicable and realistic.
2. Despite that the analysis of model throughput as a factor in OCL is intriguing, it seems that the authors did not give a perfect strategy. More analysis on how to improve the model throughput and the relationship with pre-trained models is needed.
3. Some experiments in the Appendix is better to be concluded in the main text as they also serve as some important validations on the proposed methods like results in Table6, Table11 and Table12.

**Questions:**

1. See the weakness.
2. There are some typos and grammatical errors, for example, "Plus, we also evaluate NsCE on real-world domain incremental datasets and large-scale image classification datasets." An "and" is missing here.
3. Some existing literatures like [A] have also talked about the training delay in OCL. It's better to cite it in the main text.
4. The performance improvements appear relatively marginal in terms of the last accuracy reported in Tables 4 and 5, yet there is a significant boost in performance on the AUC metric. Could there be specific reasons behind this discrepancy?

[A] Y. Ghunaim, A. Bibi, K. Alhamoud, M. Alfarra, H. A. Al Kader Hammoud, A. Prabhu, P. H.Torr, and B. Ghanem. Real-time evaluation in online continual learning: A new hope. ICCV2023

**Limitations:**

The authors have adequately addressed the limitations.

---

> ### Author Rebuttal · Authors · 2024-08-06
>
> _Respected Reviewer abk8,_ We first thank you for your valuable and insightful feedback, and for recognizing our analysis from a Pac-Bayes perspective and proposed method. Below, we address your concerns in a point-by-point manner and welcome further discussion if anything remains unclear.
>
> Q: _Discussion on more real-world scenarios where such restrictions are applicable and realistic._
> A: We agree with your perspective on the importance of realistic experimental settings and hope to address your concerns from two perspectives.
>
> 1. Evaluations on real world settings: In fact, we have included two real-world continual image classification benchmark datasets with a natural temporal evolution of visual concepts in the real world that spans a decade (2004-2014) into the experiments to include both class incremental and domain incremental settings.
> 2. Real-world applications: Similar to answer to reviewer u1kw, our initial intuition for limiting request frequency to the memory buffer is based on practical considerations. In real-world scenarios, such as autonomous vehicles and sensor network data classification where OCL could be applied, ensuring real-time accessibility of the memory buffer without incurring training delays is typically impractical.
>
> Q: _More analysis on how to improve the model throughput and the relationship with pre-trained models._
> A: To further improve model throughput, we provided a NsCE Lite version in Appendix C.2.2, which enhances throughput by not fine-tuning the entire network. We found that, in most cases, this lightweight framework achieves comparable results to NsCE, particularly on relatively simple datasets. However, we must acknowledge that such approaches heavily rely on the selection of pre-trained models. For detailed guidelines on selecting pre-trained models, please refer to Appendix C.3.
>
> Q: _Some typos, missing citations and experimental results in main text._
> A: We feel sorry for any confusion or inconvenience for you. We will cite the mentioned paper in the main text and carefully proof-read our text to ensure that no grammar mistakes or typos still exist.
>
> Q: Reasons behind the discrepancy between $A_{AUC}$ and Last accuracy.
> A: It should be noted that $A_{AUC}$ and last accuracy actually reflect different aspects of the model's performance. $A_{AUC}$ mainly assesses the real-time performance of the model, while the last accuracy measurement reflects the model's performance after processing the entire data stream. Our focus is on achieving a high-performance, anytime inference OCL model with minimal time and memory costs. Therefore, last accuracy is not our first priority. But even without employing data augmentation and knowledge distillation, our NsCE framework still achieves comparable results in last accuracy with SOTA OCL methods.

---

### Official Review · Reviewer_u1kw · 2024-07-08

**Soundness:** 4
**Presentation:** 4
**Contribution:** 3
**Rating:** 7
**Confidence:** 5

**Summary:**

This paper identifies two previously overlooked challenges in online continual learning (CL): model ignorance and myopia. In response, it introduces a new framework called Non-sparse Classifier Evolution (NsCE). NsCE features non-sparse maximum separation regularization and targeted experience replay techniques designed to quickly learn new globally discriminative features. Experimental results show significant enhancements in both model performance and throughput.

**Strengths:**

1. The paper is well-motivated by the shortcomings of existing OCL methods. It tackles a significant learning problem, and the focus on model throughput from both empirical and theoretical perspectives is a pertinent challenge in many practical applications involving data streams.
2. The strength of the paper lies in its thorough empirical evaluation, which convincingly illustrates the concepts of model ignorance and myopia. Additionally, the introduction of the term "myopia" is intriguing, and the accompanying theoretical analysis offers convincing insights into this issue.
3. The analysis of model parameter sparsity is interesting, and the proposed method with pre-trained initialization aims to address the issues of model ignorance and myopia, ultimately achieving good empirical performance.
4. It's the first time theoretical results have included discussions of model throughput, which represents a noteworthy contribution.

**Weaknesses:**

1. For the model's ignorance, following the general expectation of transfer learning that pre-trained models facilitate fast learning, it is not surprising that using pre-trained models improves the performance of OCL. So how the pre-trained model improve the learning speed is still not clear
2. As highlighted in Section 4.1 of the article, many existing methods struggle with large-scale volatile datasets. While employing pre-trained models can partially mitigate the issue of model ignorance, it is still uncertain whether this approach offers a definitive solution. Although Appendix B.2 addresses this concern, a definitive and clear solution to fully resolve this problem has not yet been established.
3. It's better to illustrate why to claim that using $max()$ function is easily affected by a small number of outliers in the parameters.
4. The paper claims that a smooth classifier can help mitigate the model's myopia, but it hampers the model's ability to perform rapid classification on the current task. Are there any insights into why this occurs?

**Questions:**

1. Is there any insight on how the size of the memory buffer relates to the number of learning tasks? The size of memory buffers seem quite random across the datasets.
2. Does performance depend on the order of the tasks/domains? Sometimes it will influence the results as different data distributions come from different tasks.
3. Is there a concern about over confidence or under confidence with this approach? Given that modern deep neural networks are particularly prone to issues with confidence levels, an analysis of this aspect would be valuable.
4. See the weakness.

---

> ### Author Rebuttal · Authors · 2024-08-06
>
> _Respected Reviewer u1kw,_ We first thank you for your valuable and insightful feedback, and for recognizing our empirical evaluation and theoretical insights. Below, we address your concerns in a point-by-point manner and welcome further discussion if anything remains unclear.
>
> Q: _How the pre-trained model improve the learning speed_
> A: We first appreciate your insightful question. As stated in Section 2, we highlighted that there is a significant trade-off between effective learning and model throughput. Meanwhile, Figure 1 clearly shows that a proper pre-trained initialization enables the model to rapidly achieve good performance on continual tasks. This, in turn, reduces the need for extensive training steps to mitigate ignorance, thereby increasing the learning speed. Certainly, fully exploiting pre-trained models to improve learning speed in OCL is a very important topic, and we hope to address this in future works.
>
> Q: _A clear solution on how to resolve the issue of model ignorance when using pre-trained models._
> A: First, we must acknowledge that, up until now, using an appropriate pre-trained initialization is the only efficient way we have found to resolve ignorance. However, our research also indicates that the efficacy of pre-training is influenced by various factors, including the domain alignment between the pre-training and target tasks, the volume and quality of the pre-training dataset, specific attributes of the target dataset, and the architecture of the backbone network. Generally, we posit that having insight into the expected data distribution of upcoming tasks allows for the selection of a pre-trained model trained on a similar distribution, which is a prudent and dependable approach.
>
> Q: _Why to claim that using $max()$ function is easily affected by a small number of outliers in the parameters._
> A: Thanks for your suggestion. By definition, the max() function selects the highest value in the dataset. It naturally does not consider the frequency or distribution of other values, which may cause some unexpected problems in optimization or evaluations.
>
> Q: _Why a smoother classifier sometimes hampers the model's ability to perform rapid classification on the current task._
> A: We have observed this phenomenon in the sensitivity analysis and ablation study. The reasons behind it are straightforward: compared to a sparse classifier, a smoother one tends to give vague predictions, which often indicate degrading performance. Therefore, it is important to ensure that $\mathcal{L}_{ce}$ remains the dominant term in optimization, while $\mathcal{L}_s$ serves more as a regularization.
>
> Q: _How the size of the memory buffer relates to the number of learning tasks?_
> A: Since each dataset comprises a varying number of classes, it is typically the case that datasets with more classes require the storage of more data, leading to a need for a larger memory buffer. Our experimental setup largely adheres to the parameters established by [1].
>
> [1] Wei Y, Ye J, Huang Z, et al. Online prototype learning for online continual learning. ICCV, 2023.
>
> Q: _Is there a concern about over confidence or under confidence with this approach?_
> A: We appreciate your insightful question and agree that the overconfidence or underconfidence of a model's predictions is a promising aspect for understanding the model's behavior during continual learning, especially in online scenarios. In fact, model myopia, recency bias in the classifier, and even ignorance can be considered from the perspective of the model's prediction confidence. For example, the bias towards the current task can be perceived from the perspective of overconfidence, and the $\mathcal{L}_s$ we proposed can help alleviate this overconfidence. However, how to evaluate the confidence level and how existing techniques addressing issues like overconfidence will perform in OCL remains unclear and warrants further exploration. We hope to leave it our future works.

---

### Official Review · Reviewer_WX6u · 2024-07-10

**Soundness:** 4
**Presentation:** 3
**Contribution:** 3
**Rating:** 8
**Confidence:** 4

**Summary:**

The paper identifies and formalizes main challenges specific to OCL. Notably the authors highlight the need for a stronger focus on ignorance (the inability of the online learner to fully converge) and throughput. Similarly, the authors identify Myopia, which corresponds to learning sparse feature, as a potential issue. Therefore they propose a simple yet effective non-sparse regularization strategy, combined with a maximum separation and targeted experience replay strategy to solve myopia with minimal computation overhead. A comparison with state-of-the-art approaches and pre-trained models shows that the proposed approach outperforms existing methods.

**Strengths:**

- The experiments seem to have been realized with care
- The paper is well written
- The experimental results are compelling
- I appreciate the defined challenges as I would also agree that Ignorance is an important topic in OCL as the focus is shifting away from forgetting in recent studies
- I appreciate the effort of providing a theoretical analysis
- the code is available

**Weaknesses:**

**Major Weaknesses**

1. What is the justification behind this limited request on the memory buffer? Why not include experiments in the traditional setup with $Freq=1$?
2. Some references to related work on budget CL are missing. I would suggest the authors to clarify what is the difference between their analysis with regard to the throughput and the analysis of computationally budgeted continual learning [1]. It would be beneficial to include such methods in the comparison table.
3. What is the justification behind the currently used metric $A_{AUC}$? I would like the authors to report the Average Accuracy, which should be the metric of interested in continual learning.
4. A graph showing the weight sparsity with and without the sparsity-regularization term could also showcase better the impact of the loss. At the moment, there is no clear demonstration of how much myopia has been solved. Overall the presentation of how the proposed approach solves the introduced challenges could be improved.
5. The authors should clarify the relation between the defined challenges and the usual stability-plasticity trade-off challenges of continual learning [6]. What is the relation between ignorance and plasticity? Is Myopia related to stability?
6. Figure captions must be enlarges. Confusion matrices are also barely readable.

**Minor Weaknesses**

7. I believe the findings of figure 2 right hand side have been discussed in previous studies such as [5]. Such references could be included in the discussion.
8. Table 1 readability could also be improved (number size specifically).
9. figure 2, right hand side has incorrect markers.
10. If I understand correctly, the "single task setting" is just training for one epoch. If so, I would advise to introduce it as such.
11. equation 3 should be reference in figure4 caption.
12. l184 : "This trend towards simplification is illustrated in Figure 4(Right), where there is a noticeable increase in the sparsity of parameters associated with older tasks as new ones are introduced." You might want to define the sparsity as $1-s(w)$ as current definition can be misleading. A high sparsity is obtained with a low $s(w)$ value.
13. I wonder how would the sparsity be affected by methods such as SS-IL [2], ER-ACE [3] or GSA [4] , which focus on re-arranged last layer weight updates.

**Typos**

- caption of Figure 4 : "class 0 corresponding to class0"
- l167: Appednix

**References**

[1] Prabhu, Ameya, et al. "Computationally budgeted continual learning: What does matter?." Proceedings of the IEEE/CVF Conference on Computer Vision and Pattern Recognition. 2023.

[2] Ahn, Hongjoon, et al. "Ss-il: Separated softmax for incremental learning." Proceedings of the IEEE/CVF International conference on computer vision. 2021.

[3] Caccia, Lucas, et al. "Reducing representation drift in online continual learning." arXiv preprint arXiv:2104.05025 1.3 (2021).

[4] Guo, Yiduo, Bing Liu, and Dongyan Zhao. "Dealing with cross-task class discrimination in online continual learning." Proceedings of the IEEE/CVF Conference on Computer Vision and Pattern Recognition. 2023.

[5] Buzzega, Pietro, et al. "Rethinking experience replay: a bag of tricks for continual learning." 2020 25th International Conference on Pattern Recognition (ICPR). IEEE, 2021.

[6] Wang, Maorong, et al. "Improving Plasticity in Online Continual Learning via Collaborative Learning." Proceedings of the IEEE/CVF Conference on Computer Vision and Pattern Recognition. 2024.

**Questions:**

See weaknesses.

**Limitations:**

The limitations have been correctly discussed in appendix.

---

> ### Author Rebuttal · Authors · 2024-08-06
>
> _Respected Reviewer WX6u,_ We first thank you for your valuable and insightful feedback, and for recognizing our motivation and theoretical analysis. Below, we address your concerns in a point-by-point manner and welcome further discussion if anything remains unclear.
>
> Q: _Justification behind the limitation on the request frequency on the memory buffer_
> A: We address this question from three aspects.
> 1. Our initial intuition for limiting request frequency to the memory buffer is based on practical considerations. In real-world scenarios, such as autonomous vehicles and sensor network data classification where OCL could be applied, ensuring real-time accessibility of the memory buffer without incurring training delays is typically impractical. This is detailed in Appendix D.1.
> 2. When take the traditional setting ($Freq=1$ and replay a small part of data in the memory each time step), the training time is typically much longer and significantly reduce model's throughput as illustrated in Figure 9 & 10, let alone replaying the whole memory buffer at $Freq=1$.
> 3. Additionally, we find that using a proper pre-trained initialization and setting $Freq=1$ enables all baselines to achieve very high performance as shown in the following table. A high-frequency replay on a moderately sized memory buffer naturally resolves most issues like forgetting, myopia and even ignorance because it makes the setting more akin to an offline, non-continual one. This, to some extent, contradicts the intuition of the OCL setting in my opinion. Here, we take the performance $A_{AUC}$ on EuroSat as an example, for each time step we replay $10$ percent of the data in memory buffer:
>
> |     | M=0.1K | M=0.2K | M=0.5K | M=1K |
> |  ----  | ----  |  ----  | ----  | ----  |
> | ER        |86.4|89.5|89.8|91.3|
> | OnPro |84.6|87.6|90.0|90.9|
> | NsCE w/o target replay|87.8|89.4|90.4|91.0|
>
> Q: _Comparison with computationally budgeted continual learning_
> A: We address this question from three aspects.
> 1. _Some missing references:_ We first thank the reviewer for pointing out the missing related work on budgeted continual learning. We will cite those references in the revised version.
>
> 2. _Difference between our analysis and theirs:_ The main difference is that in our setting, data is strictly treated as an online stream, meaning no revisit is allowed except for the data in the memory buffer. The starting points of the two papers are different: they aim to address the CL problem with limited training iterations, whereas we strive to train a model to match the original speed of the input data stream. Notably, in our experiments, the computational budget is much lower than the mentioned paper. In our setting ($Freq=1/50 or Freq=1/100$), the computational budget $\mathcal{C}$ allows revisiting less than $0.1$ percent of all observed data at given step (in average), whereas in their paper, this proportion is $25-50$ percent.
>
> 3. _Including computationally budgeted continual learning methods in the comparison table:_ We take the suggestion and add some baseline methods in computationally budgeted continual learning denoted as CBCL with ACE (We find the implementation of CBCL is very close to ER). For results for larger datasets like CLEAR and ImageNet, we will leave them in the future version. It appears that the method performs slightly better than ER in our original Table 1, but it is less competitive compared to our method.
>
> |     | CIFAR10 M=0.1K Freq=1/100 | CIFAR100 M=0.5K Freq=1/50 | EuroSat M=0.1K Freq=1/100 |
> |  ----  | ----  |  ----  | ----  |
> | CBCL w/ ACE       |84.0 $\pm$ 1.0|62.4 $\pm$ 0.8|60.9 $\pm$ 0.6|
> |     | CIFAR10 M=0.5K Freq=1/10 | CIFAR100 M=2K Freq=1/10 | EuroSat M=0.5K Freq=1/10 |
> | CBCL w/ ACE       |89.2 $\pm$ 0.4|72.7 $\pm$ 1.2|85.1 $\pm$ 0.8|
>
> Q: _The relation between the defined challenges and the usual stability-plasticity trade-off_
> The stability-plasticity trade-off is well-known in continual learning. Techniques to prevent forgetting often limit new knowledge acquisition. We thank the reviewer for giving us chances to clarify relationships between ignorance vs. plasticity and myopia vs. stability.
>
> 1. We first acknowledge that both ignorance and plasticity emphasize the need to improve the model’s capability to acquire new knowledge. However, our concept of ignorance extends beyond plasticity by incorporating the model's throughput. As illustrated in Figure 2 and Figure 11, although the method in [6], referred to as CCL or Distillation Chain in our work, enhances plasticity for OCL, it significantly reduces the model's throughput. This reduction potentially harms its practicality and performance, as detailed in Section 2.
> 2. Our examination of model myopia aims to refine the concept beyond its interpretation in the context of stability. Previous literature focusing on model stability attributes the model's decreasing performance to the interference or overlap in the feature representations or gradient contention of new and old classes, advocating for safeguarding existing knowledge to mitigate forgetting. Conversely, we contend that this performance degradation arises more from the model's limited exposure to subsets of classes during training, which causes it to favor features unique to these subsets. This leads to a myopic classification perspective with poor generalization. I think as we mentioned in our theoretical analysis (Section 4), our proposed concept of model's myopia actually offers a fresh perspective.
>
> Overall, our defined new challenges serve as an extension or an independent aspect (the other side of the coin) compared to the traditional stability-plasticity trade-off.
>
> **Note: Due to strict space limits, our responses to the other review questions are consolidated in the Author Rebuttal above. We apologize for any confusion caused by not responding individually. Please refer to the Author Rebuttal or let us know if you have any other questions to discuss further.**

---

> ### Author Response · Authors · 2024-08-07
> **Additional response**
>
> Q: _How the proposed approach solves the introduced Myopia  could be improved_
> A: We first appreciate your insightful question and suggestion to create a graph showing the weight sparsity with and without the sparsity-regularization term. We have included this in Figure 1 of the attached PDF. It can be seen that the proposed sparse regularization term $\mathcal{L}_s$ prevents the classifier from becoming excessively sparse during training. Additionally, we want to clarify that myopia is addressed by the overall design of our proposed method including the sparsity-regularization term, NsCE, as illustrated in Figure 5 of our main text. From the visualization, it can be seen that our model quickly learns the current task while minimizing confusion between past categories and those in the current task.
>
> Q: _The relation between the defined challenges and the usual stability-plasticity trade-off_
> A: The stability-plasticity trade-off is one of the most well-known concepts in continual learning. It is recognized that techniques to mitigate catastrophic forgetting often constrain a model’s capability to acquire new knowledge. Some recent studies also perceive these as independent challenges, as the reviewer mentioned in [6]. We thank the reviewer for giving us the opportunity to further clarify the relationship and differences between ignorance vs. plasticity and myopia vs. stability.
> 1. ignorance vs. plasticity: We first acknowledge that both ignorance and plasticity emphasize the need to improve the model’s capability to acquire new knowledge. However, our concept of ignorance extends beyond plasticity by incorporating the model's throughput, a critical factor in high-speed data stream environments. We observe that the single-pass nature of OCL challenges models to learn effective features within constrained training time and storage capacity, leading to a trade-off between effective learning and model throughput. As illustrated in Figure 2 and Figure 11, although the method in [6], referred to as CCL or Distillation Chain in our work, enhances plasticity for OCL, it significantly reduces the model's throughput. This reduction potentially harms its practicality and performance, as detailed in Section 2.
> 2. myopia vs. stability: Our examination of model myopia aims to refine the concept beyond its interpretation in the context of stability. Previous literature focusing on model stability attributes the model's decreasing performance to the interference or overlap in the feature representations or gradient contention of new and old classes, advocating for safeguarding existing knowledge to mitigate forgetting. Conversely, we contend that this performance degradation arises more from the model's limited exposure to subsets of classes during training, which causes it to favor features unique to these subsets. This leads to a myopic classification perspective with poor generalization. I think as we mentioned in our theoretical analysis (Section 4), our proposed concept of **model's myopia** actually offers a fresh perspective to understand the performance degradation in OCL.
>
> Overall, our defined new challenges serve as an extension or an independent aspect (the other side of the coin) compared to the traditional stability-plasticity trade-off.
>
> [6] Wang M, Michel N, Xiao L, et al. Improving Plasticity in Online Continual Learning via Collaborative Learning. CVPR, 2024.
>
> Q: _Whether the sparsity would be affected by methods that focus on re-arranged last layer weight updates_
>
> A: After the reviewer's reminder, we are also very interested in how sparsity would be affected by methods focusing on re-arranged last layer weight updates. After implementing ER-ACE and ER-AML, we found that the phenomenon of parameters rapidly becoming sparse is indeed somewhat mitigated, though not as significantly as with our proposed regularization term $\mathcal{L}_s$, as illustrated in Figure 1 of the attached PDF. Additionally, incorporating ACE or AML can also boost performance for baselines like ER and SCR. We believe that when ACE and AML nudge the learned representations to be more robust to new future classes, they indirectly decrease the sparsity of the model parameters.
>
> For GSA, the sparsity is not affected. Due to very limited time, we are not entirely sure whether this part is perfectly embedded or if further tuning would help, as the authors only provide hyperparameters for CIFAR-100 and there are some differences between the original paper's general settings and ours. For SS-IL, we did not find its implementation, so it may be left for future works.
>
> Overall, re-arranging the latter layer weight updates is an interesting and important problem in the area of OCL. We genuinely appreciate this insightful suggestion and plan to conduct detailed exploration of the dynamics of latter layer weight parameters during training in future works. This may lead to more intuitive designs.

---

> ### Author Response · Authors · 2024-08-07
> **Additional response**
>
> Q: Justification behind $A_{AUC}$ (Area under the Accuracy Curve) but not Average Accuracy $A_{avg}$
> A: At this point, we respectfully disagree with the reviewer's opinion. Compared to $A_{avg}$, $A_{AUC}$ is typically perceived as a more suitable and modern evaluation metric for the OCL scenario, especially in boundary-free settings like ours. Specifically, $A_{AUC}$ was first proposed by [1] and has been widely adopted as a crucial metric in existing OCL literature to substitute the old common used $A_{avg}$, such as [2], [3], [4]. This metric helps address the limitation of $A_{avg}$, which only provides evaluation about the model's performance at specific task transition points, usually occurring only 5-10 times in most OCL setups.
>
> In contrast to [1], we measure accuracy more frequently—at least 20 times—by evaluating it after every $\Delta n$ samples ($\Delta n=500$ for eurosat). Instead of taking an average on these observed accuracy, we compute the area under the accuracy-to-number-of-samples curve (AUC) $A_{AUC}=\frac{1}{n}\sum_{i=1}^t acc(i \cdot \Delta n) \cdot \Delta n$ (actually similar to the average taken on more observations $A_{avg}=\frac{1}{t}\sum_{i=1}^tacc(i\cdot \Delta n)$), providing a more precise evaluation than simply taking average accuracy ($n$ denotes the total number of training data). As illustrated by the following table (we directly copy the performance of RM[5] in [1]), a higher $A_{AUC}$ typically induces higher $A_{avg}$ and good real-time inference performance, but a high $A_{avg}$ does not necessarily represent good real-time inference performance. We highly respect reviewer's suggestion and plan to include a more detailed discussion including the comparison between these two metrics (including both numerical results and illustrations) in our revised version.
>
> |     | CIFAR10 | CIFAR100 | TinyImageNet | ImageNet |
> |  ----  | ----  |  ----  | ----  |----  |
> | $A_{AUC}$ |23.00 $\pm$ 1.43|8.63 $\pm$ 0.19|5.74 $\pm$ 0.30|6.22|
> | $A_{avg}$ |61.52 $\pm$ 3.69 |33.27$\pm$ 1.59|17.04 $\pm$ 0.77|28.30|
>
>
> [1] H. Koh, D. Kim, J.-W. Ha, and J. Choi. Online continual learning on class incremental blurry task configuration with anytime inference. ICLR, 2022.
> [2] Moon J Y, Park K H, Kim J U, et al. Online class incremental learning on stochastic blurry task boundary via mask and visual prompt tuning ICCV 2023.
> [3] Koh H, Seo M, Bang J, et al. Online boundary-free continual learning by scheduled data prior. ICLR, 2023.
> [4] Seo M, Koh H, Jeung W, et al. Learning Equi-angular Representations for Online Continual Learning. CVPR, 2024.
> [5]Jihwan Bang, Heesu Kim, YoungJoon Yoo, Jung-Woo Ha, and Jonghyun Choi. Rainbow memory: Continual learning with a memory of diverse samples. CVPR, 2021
>
> Q: _Readability and typos (especially on figure 2 and table 1)_
> A: We feel sorry for any confusion or inconvenience for you. We take your suggestions on the readability of figures. We will enlarge the captions of our figures to ensure a better readability. Plus, we will carefully proof-read our text to ensure that no grammar mistakes or typos still exist.

---

> ### Comment · Reviewer_WX6u · 2024-08-10
> **Thank you**
>
> I really appreciate the time and effort the authors have put into their work and rebuttal. I genuinely found this discussion very interesting and I believe this work to be of high quality.
>
> **About the $Freq=1$ scenario**
>
> I appreciate the authors honesty. My intuition was indeed that the proposed work performance gain might not be as significant in this setup. That being said, I fully agree that limiting the frequency of replay makes perfect sense and should be a prior focus compared to the $Freq=1$ scenario.
>
> **Comparison with BudgetCL**
>
> I agree with the authors comments. Thank you for the clarification.
>
> **Discussion on the link with stability-plasticity**
>
> Thank you for the clarifications.
>
> **Extra experiments**
>
> Thank you for sharing your findings. I also agree with your interpretation. I found these experiments particularly interesting and I believe them to be valuable to the community. I think such experiments would be worth including in the main draft.
>
> **Sparsity metric**
>
> Thank you for defining $\frac{1}{s(w)}$ which I believe makes more sense to me and is more easily understable. I apologize for suggesting $1-s(w)$ which of course was not suited.
>
> **On the usage of $A_{AUC}$**
>
> Thank you for the interesting reference. I still believe that the *final* average accuracy is valuable and could be reported in appendix, and I thank the authors for including it in the rebuttal. If I am not mistaken, I believe it is not currently included. In any case, I was convinced by the authors arguments and the usage of $A_{AUC}$ now makes perfect sense to me. Again, I appreciate the effort of clarifying and justifying the choices made in this work. After carefully checking the manuscript, I realize such choices were already justified in the appendix.
>
> For all the above reasons, **I will increase my score to 8.**

---

> > ### Author Response · Authors · 2024-08-11
> >
> > Thank you for your encouraging feedback and insightful suggestions.
> >
> > Regarding the final accuracy (the model's accuracy on all seen tasks after the entire training process), we have included it in Appendix C.2.1 (Table 4 and Table 5). We apologize for any potential confusion and will highlight this in the revised version.

---

> > > ### Comment · Reviewer_WX6u · 2024-08-11
> > >
> > > I see, I had seen such tables but thought it referred to the *last task accuracy* (the model's accuracy on the last task only, after training on the entire sequence). The choice of the metric as well as the naming convention are always subject to debate in Continual Learning, which is also why I was confused. Thank you for clarifying.

---

### Author Rebuttal · Authors · 2024-08-06

Dear Reviewers,
We sincerely appreciate your time and effort in reviewing our manuscript and offering valuable suggestions. Based on some of the reviews, we provide a pdf including a figure showing the redefined weight sparsity value during the training with and without the sparsity-regularization term and displaying the impact of some methods (GSA,ACE,AML) which focus on re-arranged last layer weight updates as the comparison.

Due to the strict space limits, regarding how the proposed approach solves the introduced Myopia could be improved, whether the sparsity would be affected by methods that focus on re-arranged last layer weight updates and the evaluation metrics, we hope to provide a unified answer here.

Q: _How the proposed approach solves the introduced Myopia  could be improved_
A: We first appreciate your insightful question and suggestion to create a graph showing the weight sparsity with and without the sparsity-regularization term. We have included this in Figure 1 of the attached PDF. It can be seen that the proposed sparse regularization term $\mathcal{L}_s$ prevents the classifier from becoming excessively sparse during training. Additionally, we want to clarify that myopia is addressed by the overall design of our proposed method including the sparsity-regularization term, NsCE, as illustrated in Figure 5 of our main text. From the visualization, it can be seen that our model quickly learns the current task while minimizing confusion between past categories and those in the current task.

Q: _Whether the sparsity would be affected by methods that focus on re-arranged last layer weight updates_
A: After the reviewer's reminder, we are also very interested in how sparsity would be affected by methods focusing on re-arranged last layer weight updates. After implementing ER-ACE and ER-AML, we found that the phenomenon of parameters rapidly becoming sparse is indeed somewhat mitigated, though not as significantly as with our proposed regularization term $\mathcal{L}_s$, as illustrated in Figure 1 of the attached PDF. Additionally, incorporating ACE or AML can also boost performance for baselines like ER and SCR. We believe that when ACE and AML nudge the learned representations to be more robust to new future classes, they indirectly decrease the sparsity of the model parameters.

For GSA, the sparsity is not affected. Due to very limited time, we are not entirely sure whether this part is perfectly embedded or if further tuning would help, as the authors only provide hyperparameters for CIFAR-100. For SS-IL, we did not find its implementation, so it may be left for future works.

Overall, re-arranging the latter layer weight updates is an interesting and important problem in the area of OCL. We genuinely appreciate this insightful suggestion and plan to conduct detailed exploration of the dynamics of latter layer weight parameters during training in future works. This may lead to more intuitive designs.

Q: Justification behind $A_{AUC}$ (Area under the Accuracy Curve) but not Average Accuracy $A_{avg}$
A: At this point, we respectfully disagree with the reviewer's opinion. Compared to $A_{avg}$, $A_{AUC}$ is typically perceived as a more suitable and modern evaluation metric for the OCL scenario, especially in boundary-free settings like ours. It can be seen as a refined version of $A_{avg}$ to prevent misinterpretation on model's real-time performance. $A_{AUC}$ was first proposed by [1] and has been widely adopted as a crucial metric in existing OCL literature to substitute the old commonly used $A_{avg}$, such as [2], [3], [4]. It helps address the limitation of $A_{avg}$, which only provides evaluation about model's performance at specific task transition points, usually occurring only 5-10 times in most OCL setups.

$A_{AUC}$ measures accuracy more frequently—at least 20 times—by evaluating it after every $\Delta n$ samples ($\Delta n=500$ for EuroSat). Instead of taking an average on these observed accuracy, $A_{AUC}$ computes the area under the accuracy-to-number-of-samples curve $A_{AUC}=\frac{1}{n}\sum_{i=1}^t acc(i \cdot \Delta n) \cdot \Delta n$ (actually similar to the average taken on more observations $A_{avg}=\frac{1}{t}\sum_{i=1}^tacc(i\cdot \Delta n)$), providing a more precise evaluation than simply taking average accuracy ($n$ denotes the total number of training data). As illustrated by the following table (we directly copy the performance of RM[5] in [1]), a higher $A_{AUC}$ typically induces higher $A_{avg}$ and good real-time inference performance, but a high $A_{avg}$ does not necessarily represent good real-time inference performance. We highly respect reviewer's suggestion and plan to include a more detailed discussion including the comparison between these two metrics (including both numerical results and illustrations) in our revised version.

|     | CIFAR10 | CIFAR100 | TinyImageNet | ImageNet |
|  ----  | ----  |  ----  | ----  |----  |
| $A_{AUC}$ |23.00 $\pm$ 1.43|8.63 $\pm$ 0.19|5.74 $\pm$ 0.30|6.22|
| $A_{avg}$ |61.52 $\pm$ 3.69 |33.27$\pm$ 1.59|17.04 $\pm$ 0.77|28.30|


[1] H. Koh, D. Kim, J.-W. Ha, and J. Choi. Online continual learning on class incremental blurry task configuration with anytime inference. ICLR, 2022.
[2] Moon J Y, Park K H, Kim J U, et al. Online class incremental learning on stochastic blurry task boundary via mask and visual prompt tuning ICCV 2023.
[3] Koh H, et al. Online boundary-free continual learning by scheduled data prior. ICLR, 2023.
[4] Seo M, Koh H, Jeung W, et al. Learning Equi-angular Representations for Online Continual Learning. CVPR, 2024.
[5]Jihwan Bang, Heesu Kim, YoungJoon Yoo, Jung-Woo Ha, and Jonghyun Choi. Rainbow memory: Continual learning with a memory of diverse samples. CVPR, 2021

Q: _Definition of sparsity $s(w)$._
A: We redefine sparsity score as $1/s(w)$ so that parameters with higher sparsity will have a higher sparsity score as shown in the attached PDF.

---

### Author Response · Authors · 2024-08-12

Dear Reviewers, thank you again for your thoughtful commentary. We have tried our best to address the concerns and provided detailed responses to all your comments and questions. As the deadline of discussion period is approaching, if you have any further questions, please do not hesitate to inform us, and we would be delighted to engage in any further discussions.

---

### Decision · Program_Chairs · 2024-09-25

**Decision:**

Accept (poster)

**Comment:**

This paper identifies some key issues in online continual learning, such as model ignorance and myopia, and offers ways to mitigate these using a non-sparse classifier evolution framework which allows efficient learning of globally discriminative features.

All the reviewers appreciate the paper for the careful study of the challenging in online continual learning and there was a consensus towards acceptance. The reviewers had some questions and the authors' rebuttal seems to have addressed those and the reviewers seem satisfied with the response.

Given the unanimously positive reviews, I recommend the paper for acceptance.